

# Towards a semi-asynchronous method for hydrological modeling in climate change studies

Frédéric Talbot[1], Simon Ricard[3], Jean-Daniel Sylvain[2], Guillaume Drolet[2], Annie Poulin[1], Jean-Luc Martel[1], Richard Arsenault[1]

[1] Hydrology, Climate and Climate Change Laboratory, École de technologie supérieure, Université du Québec, Montréal, H3C 1K3, Canada
[2] Direction de la recherche forestière, Ministère des Ressources naturelles et des Forêts, Québec, G1P 3W8, Canada
[3] Sciences and Engineering, Université Laval, Québec, G1V 0A6, Canada

*Correspondence to*: Frédéric Talbot (frederic.talbot.2@ens.etsmtl.ca)

**Abstract.** This study assesses the performance of the asynchronous approach used in hydrological modeling, which stands apart from the conventional approach by calibrating streamflow distributions without relying on meteorological observations. The focus is on comparing the two methods within the context of climate change impact studies, particularly in their ability to simulate key hydroclimatic processes across catchments. The analysis, conducted across multiple catchments, including a detailed case study of the Matane catchment in Southern Quebec, explores the potential of the asynchronous method as a viable alternative for future hydrological modeling. By eliminating the dependency on meteorological observations, the asynchronous approach offers potential advantages in regions with limited or unreliable observational data, providing a more flexible tool for climate change impact assessments.

The results reveal that while the asynchronous method effectively captures the overall distribution of streamflow and preserves extreme values, it faces significant challenges in accurately representing the timing of hydrological events, particularly those related to snowmelt. This issue stems, in part, from the method's decision to work directly with the biases present in raw climate model outputs, without adjusting for the timing discrepancies in meteorological inputs. Consequently, the asynchronous approach inherits these biases, leading to timing inconsistencies and increased variability across different climate models, which raises concerns about the method's ability to reliably simulate critical hydroclimatic variables under future climate scenarios. In contrast, the conventional method, which incorporates bias correction, demonstrates greater reliability in capturing the timing and magnitude of streamflow events, making it a more robust tool for most hydrological applications.

The study also highlights the concept of equifinality, where different methods achieve similar outcomes through potentially flawed mechanisms, particularly in the case of the asynchronous method. Despite projecting changes in hydroclimatic variables similar to those of the conventional method, the asynchronous approach may do so for reasons that are not hydrologically sound, particularly in snow-dominated catchments.

While the asynchronous method shows promise in preserving streamflow extremes, its current implementation requires further refinement to improve its accuracy and reliability, particularly in how it simulates the timing of seasonal dynamics. However, as climate model simulations continue to improve and their biases are progressively reduced, the asynchronous




approach is poised to benefit significantly, enhancing its potential for more accurate and reliable future hydrological

projections. The conventional method remains the preferred choice for applications requiring hydrological simulations, but

future research should focus on developing semi-asynchronous approaches that combine the asynchronous method's strength

in preserving extremes with the conventional method's ability to handle event-specific timing.

## 1 Introduction

Climate change is one of the most significant challenges of our time, with profound implications for the Earth's hydrological

systems. Alterations in temperature and precipitation regimes affect water availability and the timing of hydrological events.

Understanding these impacts is crucial for effective water resource management and decision-making (Arsenault et al., 2013;

Calvin et al., 2023). Accurate climate change studies are essential for developing strategies to mitigate and adapt to these

changes, ensuring the sustainability of natural resources and the resilience of human and ecological systems (Milly et al.,

2005; Sivakumar, 2011).

The complex dynamics of watersheds require hydrological models capable of precisely simulating both surface and

subsurface processes (Farjad et al., 2016; T. W. Chu and A. Shirmohammadi, 2004). Accurate depiction of these processes

within hydrological models is essential for assessing the impacts of climate change (Kour *et al.*, 2016; Talbot *et al.*, 2024b).

Physically based and spatially distributed hydrological models, such as the Water Balance Simulation Model (WaSiM)

(Schulla, 2021), are particularly valuable due to their detailed representation of key processes including surface runoff,

groundwater recharge, interflow, and baseflow. These models enable accurately simulating hydroclimatic variables, which

are essential for understanding the physical processes driving water flow and distribution in a catchment (Bormann and

Elfert, 2010; Förster et al., 2017, 2018; Jasper et al., 2006; Natkhin et al., 2012). The use of physically based models like

WaSiM, which capture local heterogeneity and finer-scale processes, provides a robust framework for evaluating climate

change impacts on hydrology (Devia et al., 2015; Ludwig et al., 2009; Poulin et al., 2011) and supports stakeholders in

making decisions that are both data-driven and aligned with strategic goals.

The conventional method for evaluating climate change impacts on hydrology involves a multi-step modeling chain. This

method typically starts with the calibration of a hydrological model using observed meteorological data. Subsequently, raw

climate model outputs are corrected using techniques such as quantile mapping (Jakob Themeßl et al., 2011; Mpelasoka and

Chiew, 2009) to reduce potential biases in the observed data. The calibrated hydrological model is then driven by these bias-

corrected climate data to simulate hydrological processes over both a reference and a future period.  By comparing the

differences between these two periods, the method estimates the potential effects of climate change on hydroclimatic

variables, enabling a clearer understanding of how projected climate changes will influence key hydrological processes.

While widely used, conventional methods have several limitations. Bias correction can disrupt the physical consistency

between simulated climate variables and affect long-term climate change signals (Chen et al., 2021; Lee et al., 2019).

Advanced techniques, such as multivariate quantile mapping bias correction (MBCn) (Cannon, 2018), have been developed



to address some of these issues, offering a more nuanced approach preserving the inter-variable relationships essential for reliable hydrological modeling.

Chen et al. (2021) further highlights the challenges of maintaining the integrity of climate signals due to the nonstationarity of biases in climate model outputs over time. Their study, which compares pre-processing bias correction of climate model
outputs with post-processing corrections applied directly to hydrological model outputs, reveals that while both approaches can significantly reduce biases, they also introduce uncertainties, particularly when dealing with sharp seasonal gradients in correction factors. Despite these challenges, they recommend pre-processing as the preferred method for climate impact studies. Additionally, conventional methods rely on high-quality meteorological observations, which are often unavailable in many regions (Ricard et al., 2023).
New approaches like asynchronous method have been proposed to address some of these challenges (Ricard et al., 2019, 2020; Ricard et al., 2023; Valencia Giraldo et al., 2023). This framework avoids the need for bias correction by adapting the hydrological model calibration process to directly use raw climate model projections data. This allows to conduct climate change studies without relying on observed meteorological data. Because the sequence of climatic events within climate model simulations is different from the historical observations, one cannot use the correlation between observed and
simulated streamflow during the calibration process. Instead, the asynchronous method focuses on calibrating proxies for the distribution of streamflow rather than reproducing historical time series. Given that most of climate change impact studies assess the projected change in statistical properties between a reference and a future period (Piani et al., 2010), the need for accurate temporal correlation may become less critical (Ricard et al., 2019).

Given its potential advantages, a key question is whether the asynchronous method sacrifices the integrity of hydroclimatic
variables in its pursuit of accurately reproducing streamflow distributions. To address this, this study compares hydroclimatic variables simulated by a physically based hydrological model (WaSiM) across 10 catchments, using both the conventional and asynchronous methods for climate change impact assessments. By examining the outcomes of both methods, this research aims to evaluate the asynchronous method's capacity to reliably simulate hydrological processes within catchments. The results highlight the strengths and limitations of the asynchronous framework, offering valuable
insights for advancing hydrological modeling in climate change studies.

## 2 Methods and data

### 2.1 Study area

The study focuses on a selection of forested catchments in Southern Quebec, Canada, chosen for their varied sizes and hydrological characteristics. These catchments range in area from 549 km² to 1910 km² (Table 1), providing a diverse
representation of the region's physiographic and climatic conditions (Fig. 1). This subset was selected from catchments previously studied (Talbot et al., 2024a, b), where we have extensive knowledge of their behavior and a well-established baseline for comparison. These catchments are well-suited for hydrological modeling with WaSiM, as their natural




hydrological processes remain largely intact and are minimally influenced by human-made structures such as dams. The availability of comprehensive streamflow data further supports their suitability for this study.

The region experiences a humid continental climate, with significant seasonal variation characterized by cold, snowy winters and warm, rainy summers. The Köppen-Geiger Climate Classification designates most of this region as Dfb (Humid Continental Mild Summer Wet All Year), with a smaller northern part classified as Dfc (Subarctic with Cool Summers and Year-round Precipitation) (Beck et al., 2018).

Climatic conditions show marked seasonal variations. Winters, extending from December to February, are cold with
significant snowfall, contributing to the snowpack that influences spring runoff. Average temperatures during these months frequently drop below freezing, and snow depths can accumulate substantially, impacting streamflow upon melting.

Summers, from June to August, are characterized by warm temperatures and increased rainfall (Fig. 6). The transitional seasons of spring (March to May) and autumn (September to November) exhibit moderate temperatures and variable precipitation, playing a significant role in the hydrological cycle by contributing to groundwater recharge and streamflow
variability.

**Table 1. Physical and meteorological characteristics of the selected catchments in Southern Quebec.**

| Catchments | Area (km$^2$) | Mean Elevation (m) | Most Common Soil Type | Most Common Land Use | Annual Rainfall [a] (mm) | Annual Snowfall [a] (mm) | Annual Runoff (mm) |
|---|---|---|---|---|---|---|---|
| Bonaventure | 1910 | 356 | Sandy Loam | Coniferous forest | 753 | 446 | 675 |
| Matane | 1650 | 284 | Sandy Loam | Mixed forest | 789 | 480 | 722 |
| Ouelle | 795 | 315 | Sandy Loam | Mixed forest | 826 | 424 | 604 |
| Bécancour | 919 | 305 | Sandy Loam | Deciduous forest | 1011 | 389 | 743 |
| Nicolet S-O | 549 | 259 | Sandy Loam | Cropland | 1057 | 341 | 719 |
| Au Saumon | 738 | 465 | Loam | Deciduous forest | 935 | 446 | 810 |
| Bras du Nord | 642 | 511 | Sandy Loam | Mixed forest | 1034 | 444 | 952 |
| Du Loup | 774 | 381 | Sandy Loam | Mixed forest | 795 | 355 | 504 |
| Valin | 746 | 441 | Sandy Loam | Mixed forest | 922 | 436 | 988 |
| Godbout | 1570 | 302 | Sandy Loam | Coniferous forest | 732 | 434 | 822 |

[a] Derived from WaSiM simulations for the period 1981 to 2020, using ERA5 data as input.



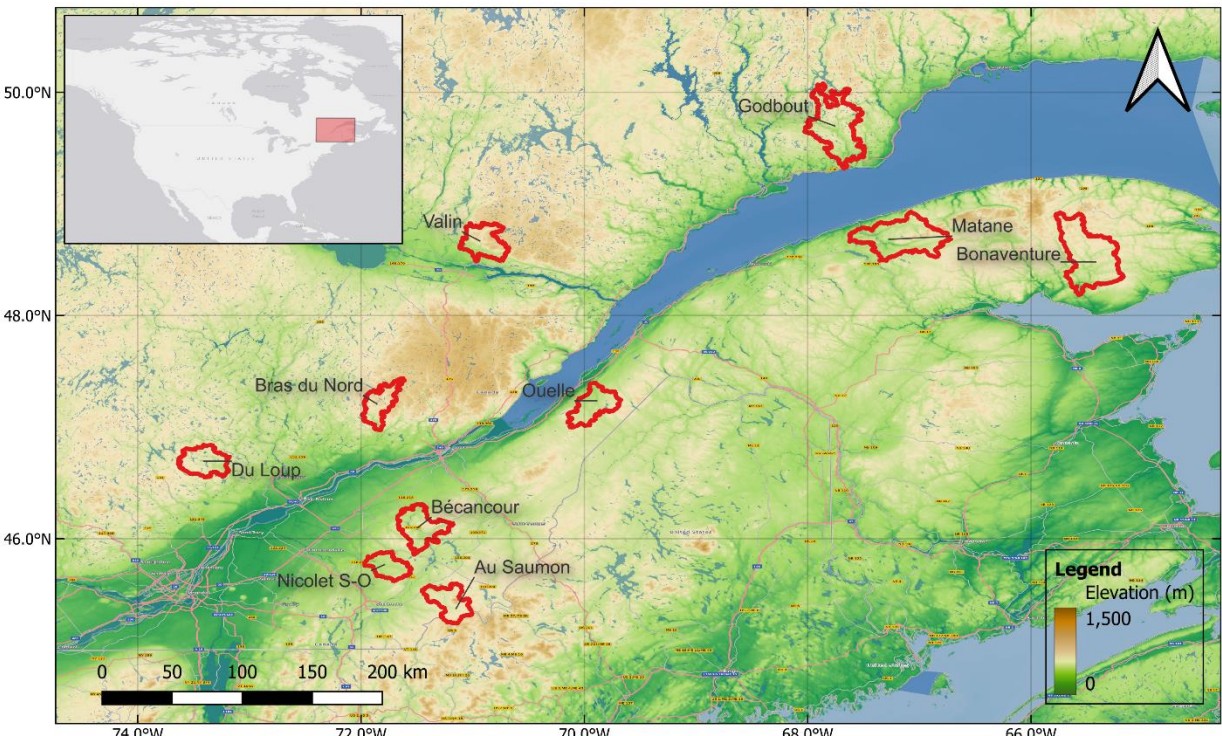

**Figure 1. Locations of the selected catchments within Southern Quebec, outlined in red. The inset map provides the location of the study area within North America.**

## 2.2 Data

### 2.2.1 Hydrometeorological

This study utilizes daily total precipitation and mean temperature data from the ECMWF Reanalysis v5 (ERA5) (Hersbach et al., 2020) for the period 1981 to 2020. ERA5 was chosen due to its advanced features over previous reanalysis datasets, such as finer spatial resolution, hourly time step, and a more sophisticated data assimilation system that incorporates a wider range of observational inputs (Tarek et al., 2020). These features make ERA5 a suitable reference dataset for hydrological modeling, as demonstrated in Tarek *et al.* (2020), where ERA5-based hydrological simulations performed equivalently to observational data in most regions, including our study area. Additionally, ERA5 showed reduced biases in temperature and precipitation compared to the ERA-Interim dataset, further justifying its use as a reliable and accurate source of climate data. Streamflow data was sourced from the *Hydroclimatic Atlas of Southern Québec* (MDDELCC, 2022), covering the period from 1981 to 2010. This dataset provides daily measurements, though some catchments have minor gaps, primarily during winter months due to ice cover and ice jams. These periods were excluded from model calibration and analyses to maintain data accuracy.



### 2.2.2 Elevation

A hydrologically conditioned digital surface model (DEM) was derived from the NASA Shuttle Radar Topography Mission version 3.0 Global 1 arc second (SRTMGL1). Hydrological corrections ensured accurate representation of hydrological networks, with adjustments made using SAGA GIS software (Conrad et al., 2015). Basin delineation and analysis were conducted using QGIS and Tanalys software (Schulla, 2021) to extract essential topographic information for hydrological modeling.

### 2.2.3 Soil type

Soil data was sourced from the SIIGSOL 100 meters database, which provides detailed descriptions of sand, clay, and silt proportions within the soil profile (Ministère des Ressources Naturelles et des Forêts, 2022; Sylvain et al., 2021). These proportions were converted to soil texture classes based on the USDA classification system (Soil Survey Division Staff, 2017). Soil hydraulic properties were imputed from established relationships between soil texture classes and hydraulic

parameters. Elevation data was used to account for soil depth variability, classifying raster cells into deep, normal, and shallow categories based on their relative elevation as described in (Talbot et al., 2024a).

### 2.2.4 Land use

Land use data was obtained from the 2015 North American Land Change Monitoring System (NALCMS) 30 meters dataset. This data was resampled using the nearest neighbor method to create land use maps, significantly impacting hydrological

parameters such as root distribution, vegetation cover fraction (VCF), roughness length (Z0), and albedo. These parameters influence processes like evapotranspiration, runoff, and infiltration (2015 Land Cover of North America at 30 meters, 2023; Latifovic et al., 2012).

### 2.2.5 Climate models

Projected daily temperature and precipitation data were sourced from the Coupled Model Intercomparison Project Phase 6

(CMIP6) (O'Neill et al., 2016) for both the reference period (1981-2010) and future period (2070-2099). These datasets were accessed and processed through the PAVICS-Hydro platform (Arsenault et al., 2023). The Shared Socioeconomic Pathway 5-8.5 (SSP5-8.5) scenario, which projects very high greenhouse gas emissions, was used to simulate future conditions (Calvin et al., 2023). To address uncertainties related to climate model selection, an ensemble of 18 climate models was employed as it was previously shown that, to ensure robustness, using multiple climate models is required (Arsenault et al.,

2020; Lucas-Picher et al., 2021; Minville et al., 2008; Tarek et al., 2021). This ensemble approach ensures a more robust representation of potential climate outcomes by capturing a range of possible future scenarios.





### 2.3 Hydrological modelling

#### 2.3.1 Hydrological model

WaSiM is a physically based, spatially distributed hydrological model designed to simulate water flow processes in
catchments. It integrates a comprehensive suite of sub-models to capture key hydrological processes, including surface runoff, groundwater recharge, interflow, and baseflow, within a deterministic framework (Schulla, 2021).

In this study, WaSiM was configured with a spatial resolution of 1000 meters and a temporal resolution of 24 hours. This setup allows for detailed spatial analysis while maintaining computational efficiency. The chosen spatial resolution ensures that the heterogeneity of the landscape is adequately captured, and the daily time step allows for accurate simulation of
hydrological processes over time.

WaSiM employs the Richards equation and the Van Genuchten parameters for simulating water flow in the unsaturated zone (van Genuchten, 1980; Richards, 1931). This equation provides a physically based representation of hydraulic head gradients and soil moisture dynamics, incorporating detailed soil physical properties. Groundwater flow is calculated conceptually within the unsaturated zone model.

**2.3.2 Conventional method**

The framework used to calibrate and validate the effectiveness of the conventional method rely on the split sample test (SST) approach, which is widely recognized for its effectiveness in evaluating model performance. This approach involves dividing the data into separate calibration and validation periods, allowing for an assessment of the model's ability to generalize beyond the calibration conditions.

For this method, historical data from ERA5 were used for both calibration and validation. The calibration period spanned from 2000 to 2009, during which simulations were performed over a 15-year period (1995 to 2009), discarding the first 5 years to stabilize the initial conditions of the model. The validation period was set from 1990 to 1999, following the same approach of conducting simulations over a 15-year period (1985 to 1999) and discarding the initial 5 years.

A set of 17 parameters (Table 2) was selected for calibration based on model documentation and the configuration used in
Talbot *et al.* (2024a). Table 2 was taken from Talbot et al. (2024a).






**Table 2. Calibration parameters for the hydrological model WaSiM.**

| No. | Code | Description | Sub-Model | Range |
|---|---|---|---|---|
| 1 | $k_D$ | Storage coefficient for surface runoff (h) | Unsaturated zone | [1, 25] |
| 2 | $k_H$ | Storage coefficient for interflow (h) | Unsaturated zone | [1, 25] |
| 3 | $d_r$ | Drainage density for interflow (m⁻¹) | Unsaturated zone | [1, 50] |
| 4 | $QD_{Snow}$ | Fraction of surface runoff on snow melt | Unsaturated zone | [0.1, 1] |
| 5 | $c_0$ | Degree-Day factor (mm °C⁻¹ d⁻¹) | Snow | [0, 3] |
| 6 | $T_0$ | Temperature limit for snow melt (°C) | Snow | [-4, 4] |
| 7 | $T_{R/S}$ | Transition temperature snow/rain (°C) | Snow | [-4, 4] |
| 8 | $C_{WH}$ | Water storage capacity of snow | Snow | [0.1, 0.3] |
| 9 | $C_{rfr}$ | Coefficient for refreezing | Snow | [0.1, 1] |
| 10 | $f_{i,summer}$ | Summer correction factors for PET | Evapotranspiration | [0.1, 2] |
| 11 | $f_{i,fall}$ | Fall correction factors for PET | Evapotranspiration | [0.1, 2] |
| 12 | $f_{i,winter}$ | Winter correction factors for PET | Evapotranspiration | [0.1, 2] |
| 13 | $f_{i,spring}$ | Spring correction factors for PET | Evapotranspiration | [0.1, 2] |
| 14 | $K_{rec}$ | Recession constant for hydraulic conductivity | Soil table | [0.1, 0.99] |
| 15 | $d_z$[a] | Soil layer thickness | Soil table | [0.8, 1.4] |
| 16 | KB | Storage coefficient for base flow (m) | Unsaturated zone | [0.1, 8] |
| 17 | Q0 | Scaling factor for base flow (mm h⁻¹) | Unsaturated zone | [0.1, 5] |

[a] Calibration coefficient, ranging from 0.8 to 1.4, is applied to adjust the total soil depth, which is predetermined to be 8 meters for shallow, 14 meters for normal, and 20 meters for deep soil conditions.

These parameters were optimized based on a single objective function through the Dynamically Dimensioned Search (DDS) algorithm, developed by Tolson and Shoemaker (2007). This algorithm was chosen for its efficiency in handling complex

optimization problems for compute-intensive hydrological models, as recommended by Arsenault *et al.* (2014).

The objective function used for calibration was the Kling Gupta-Efficiency (KGE) (Kling et al., 2012). The KGE metric provides a balanced evaluation of model performance by considering simultaneously correlation, variability, and bias in the simulated streamflow relative to observed streamflow.

The KGE is computed using Eq. (1):

$$KGE = 1 - \sqrt{(r-1)^2 + \left(\frac{\sigma_{sim}/\mu_{sim}}{\sigma_{obs}/\mu_{obs}} - 1\right)^2 + \left(\frac{\mu_{sim}}{\mu_{obs}} - 1\right)^2}, \tag{1}$$

where $r$ is the correlation coefficient between simulated and observed streamflow, $\sigma_{sim}$ is the standard deviation of simulated streamflow, $\sigma_{obs}$ is the standard deviation of observed streamflow, $\mu_{sim}$ is the mean of the simulated streamflow, and $\mu_{obs}$ is the mean of the observed streamflow.



A KGE value of 1 indicates a perfect match between the simulated and observed streamflow, reflecting ideal performance
across all three components: correlation, variability, and bias.

To address biases in the climate model simulations, the Multivariate Bias Correction algorithm (MBCn) of Cannon (2018)
was utilized. This method corrects biases in meteorological data while accounting for spatiotemporal interdependencies
between variables and preserving changes in quantiles between the reference (1981-2010) and future (2070-2099) periods.
The bias correction was applied to daily total precipitation and daily mean temperature using ERA5 data as the reference
over the period 1981-2010 and was used to correct the climate models data for both the reference (1981-2010) and future
periods (2070-2099).

For the conventional method, climate change hydrological simulations were performed using the bias-corrected climate
models data. Simulations were conducted for each climate model over the reference and future periods using the calibrated
hydrological model for each catchment.

### 2.3.3 Asynchronous method

The primary objective of the asynchronous method is to conduct climate change studies without relying on observed
meteorological data (Ricard et al., 2019, 2020; Ricard et al., 2023) and eliminate the need for bias-correction of climate
variables. Instead, the calibration is performed using raw climate model data and observed streamflow, integrating the bias-
correction in the calibration step. A significant challenge in this approach is the lack of synchronization between the timings
of observed streamflows and those of raw climate model outputs (Ricard et al., 2019), as climate models are not temporally
aligned with actual past events. This requires a departure from the conventional calibration framework, which aims to
optimize the synchronicity and amplitude of streamflow.

To overcome this obstacle, the objective function optimizes the distribution of observed streamflow over an extended period
rather than individual streamflow observations. This approach ensures the hydrological model effectively captures the
streamflow distribution, rather than day-to-day natural variability.

Given the calibration objectives of the asynchronous method, the observed streamflow data from 1984 to 2009 was sorted
and used to establish a reference distribution of streamflow. This sorted distribution provided a consistent target for both the
calibration and validation of the hydrological model. In the context of the asynchronous method, where direct temporal
alignment between climate model outputs and observed streamflow is not maintained, relying on the same observed
distribution for both calibration and validation ensures that the model is evaluated against a stable and representative
reference. Therefore, a 25-year period (1984-2009) was used for both calibration and validation, as it captures a broad range
of hydrological conditions, minimizes the influence of short-term climate variability, and mitigates biases that could arise
from using shorter time frames. A key hypothesis underlying this approach is the assumption of stationarity—that the
hydrological model, with fixed parameters optimized during calibration, will continue to produce reasonable streamflow
simulations under future climate conditions. This assumes that despite changing climatic conditions, the model will
adequately respond to future scenarios as it did to past conditions. However, if future climate changes introduce conditions



outside the model's calibrated range, such as new snow patterns or shifts in seasonal dynamics, the model's performance could be compromised.

The calibration and validation periods are separated based on the total yearly precipitation from October to September. This separation ensures an equal distribution of wet and dry years between both periods. Simulations were performed for the years 1984 to 2011, with the first two years discarded to allow for initial model stabilization. Out of the 26 years of simulations, 13 years were used for calibration, selected based on total yearly precipitation, while all 26 years were utilized for model validation.

To address biases in simulated streamflow resulting from biases in precipitation and temperature in the raw climate data, the simulated streamflow was adjusted by multiplying it by a factor equal to the ratio of the mean observed streamflow $Q_{obs}$ to the mean simulated streamflow $Q_{sim}$. This adjustment was applied only during the calibration process and not during the reference or future periods simulations. It ensures that the mean simulated streamflow matches the mean observed streamflow, effectively removing bias. This means the simulated absolute streamflow values cannot be directly compared with observations, but changes between the reference and future period can be analyzed.

The Root Mean Square Error (RMSE) was employed as the objective function:

$$RMSE = \sqrt{\frac{1}{n}\sum_{i=1}^{n}\left(Q_{sim_i} - Q_{obs_i}\right)^2}, \qquad (2)$$

where $Q_{sim}$ represents the sorted simulated streamflow (mm), $Q_{obs}$ represents the sorted observed streamflow (mm), and $n$ is the number of simulated streamflow values.

Each climate model was calibrated for each catchment, resulting in a total of 180 calibration parameter sets (18 climate models x 10 catchments). In contrast, the conventional method involves 10 calibration parameter sets (one per catchment) which is then applied to all climate models and their bias-corrected outputs. Consequently, the asynchronous method is considerably more computationally intensive than the conventional method in terms of parameter calibration.

The calibration framework for the asynchronous method is similar to that of the conventional method. It involves 1000 trials, uses the same 17 calibration parameters (Table 2), and employs DDS optimization algorithm.

For the asynchronous method, climate change simulations were conducted using the calibrated model for each catchment and each projected climate model, but without relying on historical event timing. Raw projected climate data were utilized to perform simulations over both the reference and future periods.

## 2.4 Comparative analysis

The comparative analysis in this study is designed to evaluate the performance of the conventional and asynchronous methods in simulating hydrological processes under both current and future climate conditions. To ensure a fair and unbiased comparison, both methods employed the same WaSiM configuration, including identical calibration parameters, the number of evaluations, and the optimization algorithm. This was performed to ensure minimal calibration bias and to isolate the differences attributable solely to the methodological framework of each approach.





The first step in the comparative analysis involves assessing the calibration and validation performance of each method.
Streamflow simulations were evaluated using the KGE for the conventional method and the RMSE of the sorted simulated and observed streamflow for the asynchronous method. These metrics were selected to highlight each method's strengths in different aspects of streamflow simulation—KGE for overall model performance and RMSE for the accuracy of flow distribution.

Beyond streamflow, we also examine the relationships between various hydroclimatic variables, such as groundwater
recharge, surface runoff, soil moisture, and snow water equivalent (SWE). By comparing the simulated values from both methods, the analysis seeks to understand how well each method captures the interactions between these variables. This serves to evaluate the internal consistency of the models and assess their ability to realistically simulate the physical processes within the catchments.

The analysis extends to a comparison of the projected changes in hydroclimatic variables between the reference period
(1981–2010) and the future period (2070–2099). The magnitude and direction of these changes are assessed to determine how each method projects the impact of climate change on the catchments. This includes examining variables such as changes in snowmelt dynamics, and the consequent effects on streamflow, surface runoff and groundwater recharge.

A detailed spatial analysis is conducted to evaluate the distribution of key variables, such as soil moisture and groundwater recharge, across the catchments. The comparative analysis employs several criteria to determine which method is more
effective. These include the accuracy of streamflow simulation (both in terms of overall distribution and event timing), the internal consistency of hydroclimatic variable relationships and the realism of spatial distributions and projected changes under future climate scenarios. The method that consistently demonstrates superior performance across these criteria is considered more reliable for future hydrological modeling and climate impact assessments.

## 3 Results

### 3.1 Streamflow representation performance

For the conventional method, streamflow representation performance was assessed using the KGE metric for each catchment during both the calibration and validation periods.

During calibration, the conventional method achieves KGE values ranging from 0.817 to 0.906, with a mean of 0.863. Similarly, for the validation period, the KGE values ranged from 0.778 to 0.906, with a mean of 0.842. These results indicate
that the conventional method maintains a consistent performance in simulating streamflow across different catchments. Detailed KGE results for each catchment are provided in the Appendix A (Table A1).

For the asynchronous method, the RMSE was used to evaluate streamflow representation performance during both the calibration and validation periods. During calibration, the RMSE values exhibited a mean of 0.121 mm d$^{-1}$ with a mean standard deviation between climate models of 0.031 mm d$^{-1}$. In the validation period, the RMSE values demonstrated similar





patterns, with a mean of 0.179 mm d$^{-1}$ and a standard deviation of 0.057 mm d$^{-1}$. Detailed RMSE values are available in the Appendix A (Table A2).

Figure 2 presents hydrographs of streamflow for both methods during the reference period across the ten catchments, along with observed streamflow for the same period. The asynchronous method shows greater variability between climate models, especially in the timing of peak flows, which often fails to align with the observed data, as expected. This variability

suggests that the timing of streamflow events in the asynchronous method is highly sensitive to the specific climate model employed. For instance, in the Matane catchment, the observed and conventional method peak flow occurs at the beginning of May, while the asynchronous method shows a broader range of peak flow timings, extending from early May to late June. This discrepancy might be attributed to challenges in accurately simulating snowmelt processes with the asynchronous method, which are crucial for generating high flows in the study region. Furthermore, in the same catchment, the

asynchronous method overestimates summer flows compared to the observed data, indicating potential difficulties in capturing the seasonal dynamics of low-flow periods. Conversely, the conventional method accurately reproduces the annual observed streamflow variability, demonstrating its strength in capturing the timing and magnitude of hydrological events.

Despite these differences, the asynchronous method outperforms the conventional method in terms of annual volume accuracy in 8 out of 10 catchments (Fig. 2). This enhanced accuracy can be attributed to the scaling adjustments applied

during the calibration period. In the asynchronous method, streamflow is adjusted by multiplying them by a scaling factor to correct biases between simulated and observed means, effectively minimizing the difference between overall volume in simulated and observed streamflow. However, the inability of the asynchronous simulations to replicate the precise timing of streamflow events raises concerns about its representation of underlying hydroclimatic variables, which may impact the model's broader predictive accuracy.





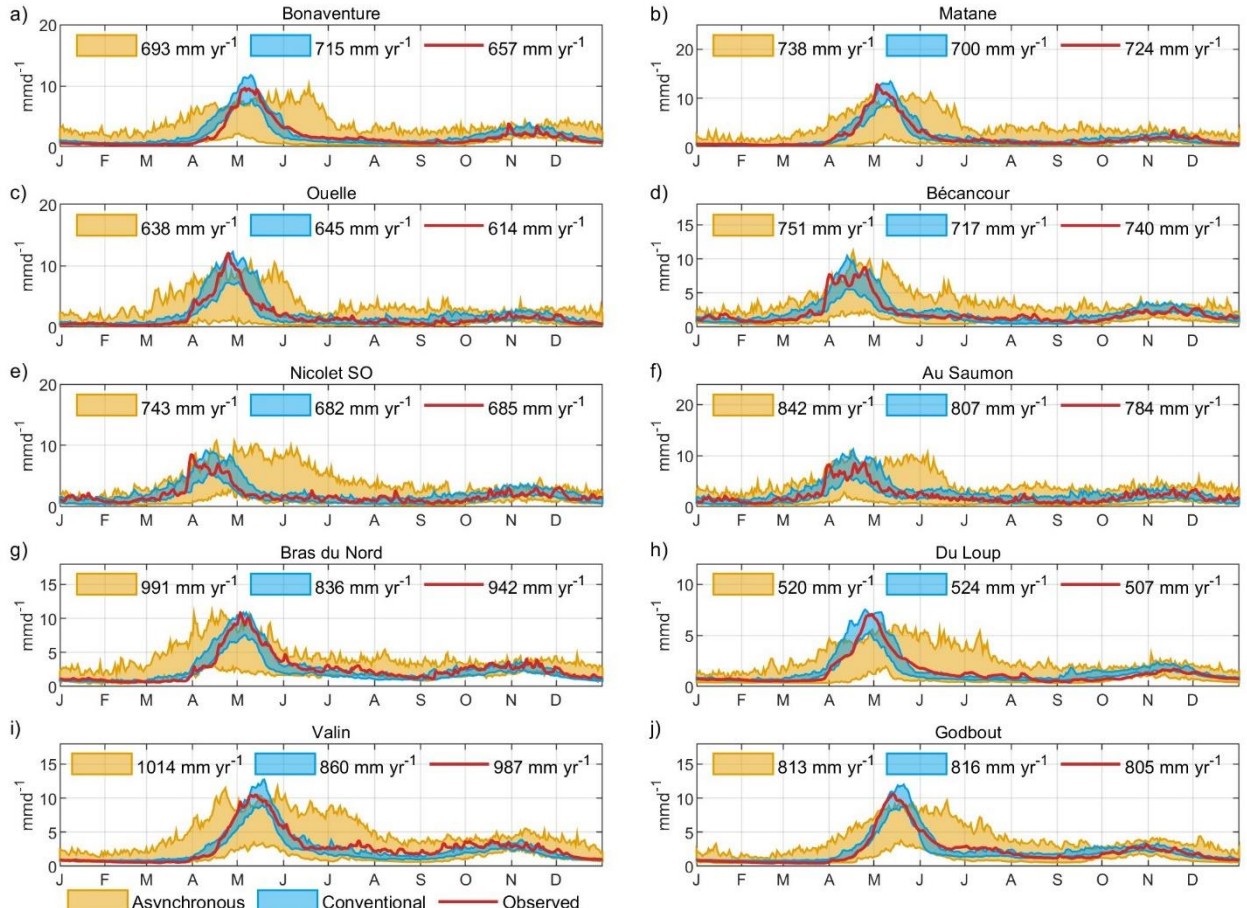

**Figure 2. Seasonal streamflow comparison between the asynchronous and conventional methods and observed data across ten catchments during the reference period (1981–2010). The panels (a-j) represent the catchments of Bonaventure, Matane, Ouelle, Bécancour, Nicolet SO, Au Saumon, Bras du Nord, Du Loup, Valin, and Godbout, respectively. The colored bands indicate the range of daily streamflow simulated by the asynchronous method (yellow) and conventional method (blue) alongside the observed streamflow (red line).**

Figure 3 shows the relationship between the sorted streamflow distribution and percentage bias for simulated and observed values using conventional and asynchronous methods during the reference period (1981–2010) for the Matane catchment. Climate models using the conventional method exhibit broader dispersion compared to the asynchronous method. Moreover, the asynchronous method demonstrates a better ability to capture extreme streamflow events, as indicated by its lower percentage bias across a range of streamflow conditions. This suggests that the asynchronous method is more effective in predicting extreme flows and capturing the overall distribution of streamflow, likely due to its calibration focus on streamflow distribution rather than the precise timing of hydrological events.





While these results are specific to the Matane catchment, similar patterns are observed across all catchments studied (Fig. B1 to Fig. B9). This consistency highlights the asynchronous method's strength in capturing the full range of potential

streamflow conditions across diverse hydrological settings, particularly in representing extremes.

In contrast, the conventional method excels at capturing the annual fluctuations and timing of the observed streamflow but is more limited in its ability to represent extreme flows. The asynchronous method, optimized for distributions, offers a distinct advantage in this area by more accurately reflecting the range of possible streamflow values, particularly during high and low flow events.

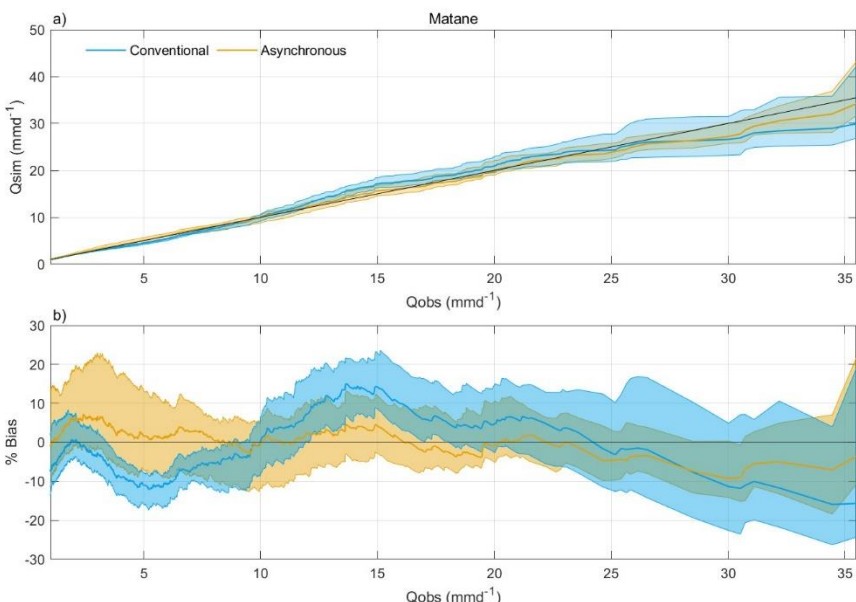


**Figure 3. Performance comparison between the conventional and asynchronous methods for the Matane catchment during the reference period (1981–2010). Panel (a) displays the relationship between simulated daily streamflow (Qsim) and observed daily streamflow (Qobs) for the conventional (blue) and asynchronous (yellow) methods. The x-axis represents the observed daily streamflow, while the y-axis represents the simulated streamflow. Panel (b) shows the percentage bias between observed and**
**simulated streamflows for both methods, with the x-axis representing the observed daily streamflow and the y-axis displaying the percentage bias relative to the observed values. The shaded regions in both panels illustrate the variability among climate models around the mean bias for each method, emphasizing the differences in how well each method simulates streamflow across the observed streamflow range.**

Figure 4 presents a comparative analysis of streamflow quantiles for both methods across the reference (1981–2010) and
future (2070–2099) periods for all ten catchments. When comparing the observed streamflow to the reference period simulations, the asynchronous method shows a closer alignment with the observed distribution, particularly for high flows (Q95% and Q90%), confirming its strength in capturing extreme events. However, it is important to note that the asynchronous method also exhibits a higher dispersion among climate models during the reference period, indicating greater variability in its predictions.

In terms of future projections, both methods demonstrate similar trends, with both projecting decreases in high flows (Q95%) across most catchments. However, the projections for other flow quantiles, such as median (Q50%) and low flows





(Q5% and Q10%), show mixed changes that vary depending on the catchment's geographical location. While there is a broad agreement between the methods on the direction of changes from the reference to the future period, the asynchronous method again shows a higher dispersion between climate models, especially for low flows (Q5% and Q10%). This increased
variability suggests that while the asynchronous method is effective at representing the distribution of streamflow, it may introduce greater uncertainty in future projections, particularly for low-flow conditions.



**Figure 4. Streamflow distribution analysis for both the reference (1981–2010) and future (2070–2099) periods across 10 catchments, comparing the conventional (blue) and asynchronous (orange) methods. The figure illustrates the percentage**
**differences in streamflow quantiles between the simulated and observed streamflow for both methods with the central line indicating the median and whiskers extending to the 25th and 75th percentiles. The panels represent the following streamflow quantiles: (a) Q95% (high flow), (b) Q90%, (c) Q50% (median flow), (d) Q10%, and (e) Q5% (low flow). For each catchment, the reference period data is shown in darker shades, while the future period projections are displayed in lighter shades. Blue represents the conventional method, and orange represents the asynchronous method.**



**3.2 Hydroclimatic variables**

The primary objective of this study is to compare the conventional and asynchronous methods and evaluate the ability of the asynchronous approach to accurately reproduce the physical processes within catchments. This section expands the analysis beyond streamflow to include a broader range of hydroclimatic variables, providing a more comprehensive assessment of the methods' performance.

Table 3 presents the annual averages for several key hydroclimatic variables—such as precipitation, snowfall, streamflow, surface runoff, interflow, actual evapotranspiration (ETa), baseflow, groundwater recharge, SWE, and soil moisture—across both the reference (1981–2010) and future (2070–2099) periods. The table also highlights the relative changes in these variables between the reference and future periods for both the conventional and asynchronous methods.

When comparing the results for the reference and future periods, both methods exhibit similar trends across most
hydroclimatic variables. For instance, both methods predict an increase in precipitation and ETa, alongside a significant reduction in snowfall and SWE as a response to the anticipated warming climate.

However, notable differences arise in the representation of certain variables. One of the most significant discrepancies is observed in surface runoff. The asynchronous method predicts more than twice the amount of surface runoff compared to the conventional method, both in the reference and future periods. This substantial difference suggests that the asynchronous
method may be simulating surface processes differently than the conventional method.

In terms of relative changes between the reference and future periods, both methods demonstrate similar trends across most variables, indicating agreement on the direction of change due to climate impacts. For example, both methods predict a similar reduction in snowfall (around 33% and 41%) and SWE (around 53% and 58%), reflecting the expected decrease in snow accumulation as temperatures rise. The increase in Eta (31%) is also consistent across both methods, suggesting that
higher temperatures will lead to greater evapotranspiration.







**Table 3. Comparison of hydroclimatic variables between the reference (1981–2010) and future (2070–2099) periods for both the conventional and asynchronous methods across 10 catchments and 18 climate models. The table presents annual averages for key hydroclimatic variables, including precipitation, snowfall, streamflow, surface runoff, interflow, actual evapotranspiration (ETa), baseflow, groundwater recharge, snow water equivalent (SWE), and soil moisture. Relative changes between the reference and future periods are also provided for each method.**

| Hydroclimatic Variables | Unit | Conventional | | | Asynchronous | | |
| --- | --- | --- | --- | --- | --- | --- | --- |
| | | Reference (1981-2010) | Future (2070-2099) | Relative Change | Reference (1981-2010) | Future (2070-2099) | Relative Change |
| Precipitation | mm yr⁻¹ | 1276 | 1463 | 15% | 1328 | 1507 | 13% |
| Snowfall | mm yr⁻¹ | 409 | 273 | -33% | 362 | 214 | -41% |
| Streamflow | mm yr⁻¹ | 730 | 744 | 2% | 771 | 778 | 1% |
| Surface Runoff | mm yr⁻¹ | 109 | 77 | -29% | 256 | 199 | -22% |
| Interflow | mm yr⁻¹ | 482 | 530 | 10% | 400 | 460 | 15% |
| ETa | mm yr⁻¹ | 554 | 727 | 31% | 561 | 733 | 31% |
| Baseflow | mm yr⁻¹ | 140 | 137 | -2% | 113 | 117 | 3% |
| Groundwater Recharge | mm yr⁻¹ | 133 | 139 | 4% | 118 | 141 | 19% |
| SWE | mm | 281 | 131 | -53% | 252 | 106 | -58% |
| Soil Moisture | - | 0.188 | 0.180 | -5% | 0.203 | 0.200 | -2% |

Figure 5 illustrates the annual variations of key hydroclimatic variables across different catchments and 18 climate models for both the reference and future periods, comparing the conventional and asynchronous methods. Several noteworthy differences emerge between the two methods, particularly in streamflow, interflow, and groundwater recharge.

The conventional method tends to produce higher peaks for streamflow, interflow and recharge during periods of high streamflow, suggesting a more pronounced response to snowmelt and precipitation events. This method also exhibits greater variability during these high-flow periods. In contrast, the asynchronous method displays higher summer flows.

Both methods predict similar absolute changes in streamflow between the reference and future periods, reflecting a consistent trend across climate projections. However, the conventional method indicates a more substantial decrease in high flows in the future, which could have significant implications for water resource management, particularly in regions where peak flows are crucial for reservoir replenishment and flood control. It is thus important to assess if one of the two methods (conventional vs. asynchronous) can be considered as more reliable than the other as this could affect conclusions of many climate change impact studies.

For surface runoff, the asynchronous method consistently generates higher values compared to the conventional method across both periods. This suggests that the asynchronous method may be simulating more rapid or intense surface processes. Despite these differences in magnitude, both methods exhibit similar trends in absolute change, indicating a projected decrease in surface runoff in the future.

ETa results are closely aligned between the two methods. This consistency suggests that ETa projections are robust across different modeling frameworks, reinforcing confidence in these projections for water balance assessments.



Maximum SWE also shows differences between the two methods. The conventional method predicts higher SWE values, suggesting that it may simulate a more substantial accumulation of snowpack during the winter months. On the other hand, the asynchronous method demonstrates greater variability during the snowmelt period, with snowmelt extending from April

to August, compared to a more concentrated snowmelt period from April to June in the conventional method. This extended snowmelt period in the asynchronous method could lead to prolonged high flows in late spring and early summer. However, it is quite unrealistic to observe snow persisting through the summer months, highlighting a significant limitation of the asynchronous method in accurately representing seasonal snow dynamics.

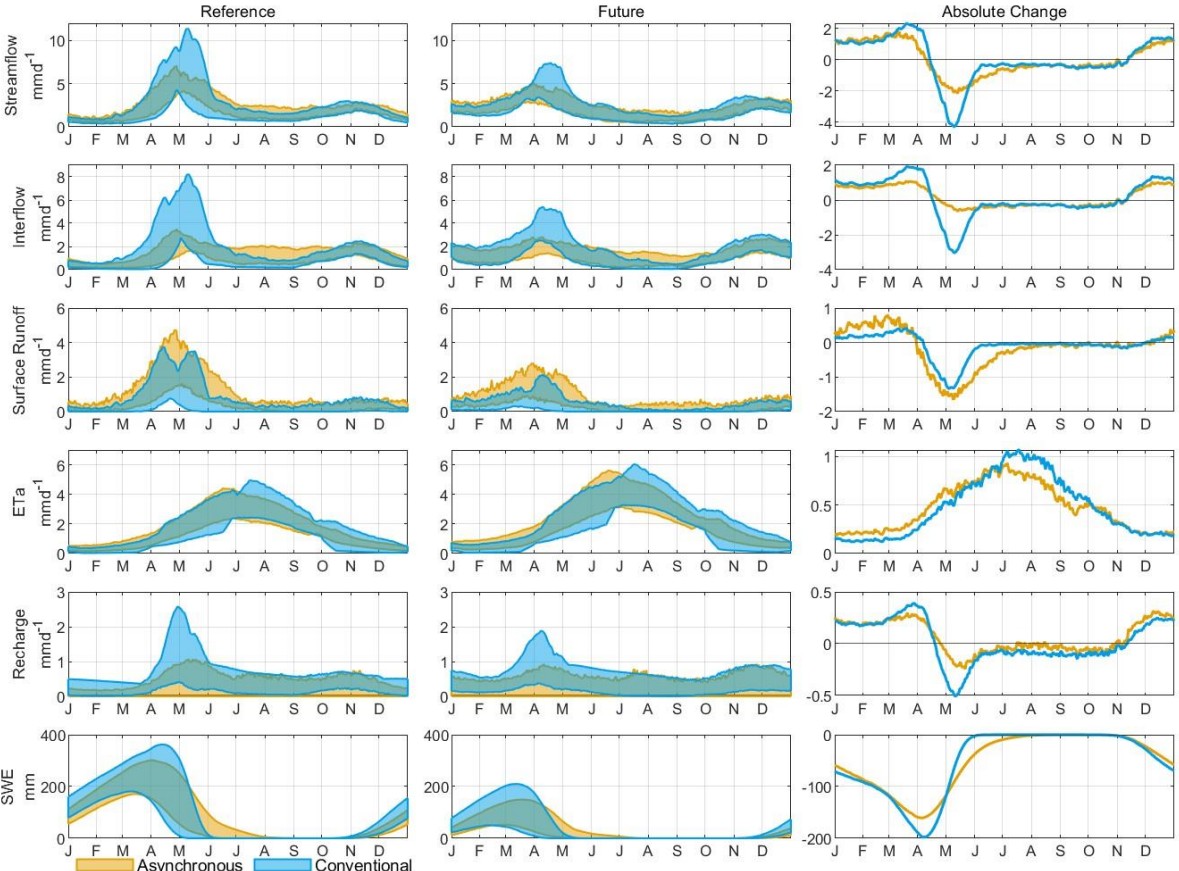

**Figure 5. Seasonal distribution of key hydroclimatic variables for the reference period (1981–2010), future period (2070–2099), and their absolute changes across the 10 catchments and 18 climate models using both the conventional and asynchronous methods. The figure presents the monthly averages of streamflow, interflow, surface runoff, actual evapotranspiration (ETa), groundwater recharge, and snow water equivalent (SWE). The left column shows the reference period, the middle column displays the future period, and the right column illustrates the absolute changes between the two periods. The shaded areas represent the**
**variability across the catchments.**



### 3.3 Case study

To thoroughly assess the asynchronous method's ability to accurately reproduce physical processes within a catchment, the Matane catchment, covering an area of 1650 km², was selected as a case study. This catchment was chosen due to its strong calibration and validation performance under both methods, as well as its representative characteristics of the broader set of
studied catchments.

Average monthly temperature and precipitation for the reference period (1981-2010) and the future period (2070-2099) before and after bias-correction using MBCn for the Matane catchment and the climate model ACCESS-ESM1-5 is provided as an example (Fig. 6).

In the reference period, a noticeable gap exists between the ERA5 data and the raw climate model data, with the raw climate
data showing higher temperatures and increased precipitation for all months except October. The effectiveness of the bias correction is evident, as the bias-corrected climate data closely aligns with the ERA5 data, significantly reducing discrepancies in temperature and precipitation. The same bias trend is observed in the future period, where raw climate model data predicts higher temperatures and increased precipitation compared to the bias-corrected data.

Furthermore, Fig. 6 highlights the anticipated changes in precipitation and temperature between the reference and future
periods. Temperatures are expected to increase significantly, with projected increases around 6 degrees Celsius across the study area. These projections are in line with the IPCC's forecasts based on SSP5-8.5, and suggest that northern latitudes will experience faster warming compared to the global average (Estrada et al., 2021). Projections consistently show an annual increase in precipitation ranging from 15 to 20%, with the most significant increases occurring between December and April, as well as in July. The anticipated increases in temperature and changes in precipitation patterns have profound implications
for hydrological processes and water resource management.





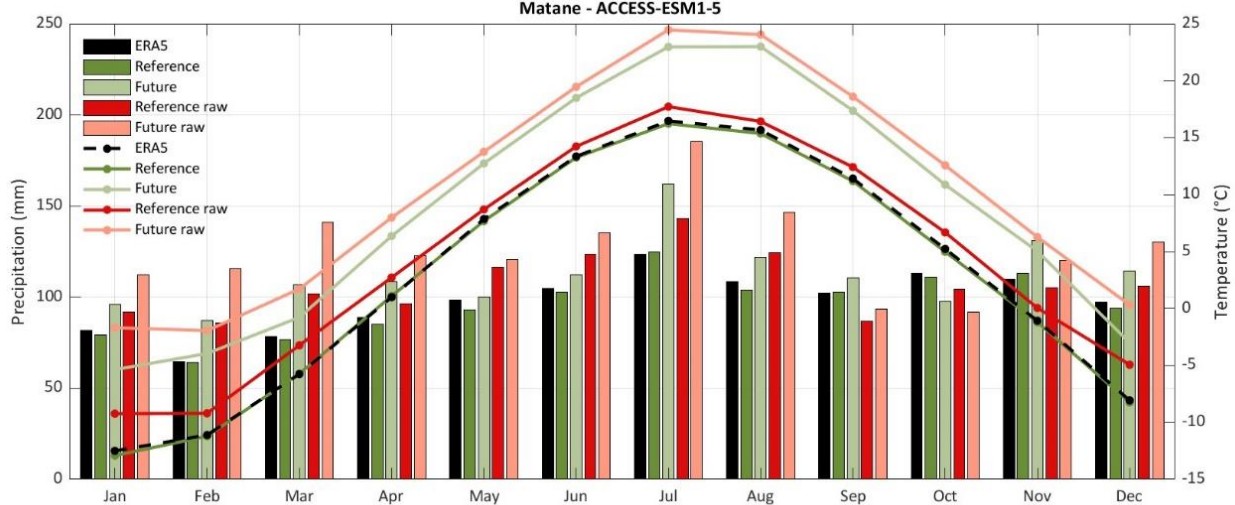

**Figure 6. Comparison of average monthly precipitation and temperature for the Matane catchment. This figure displays the average monthly temperature and precipitation for the Matane catchment during the reference period (1981-2010) and the future period (2070-2099) under the Shared Socioeconomic Pathway 5-8.5 (SSP585) scenario. Solid lines represent average monthly temperature: green for the reference period with bias correction, light green for the future period with bias correction, red for the reference period without bias correction, and light red for the future period without bias correction. The black dashed line indicates ERA5 data for comparison. Bars represent average monthly precipitation: black for ERA5 data, green for precipitation with bias correction, and red for precipitation without bias correction. Lighter shades of the bars correspond to data for the future period, distinguishing between bias-corrected and uncorrected scenarios.**

Figure 7 offers a detailed comparison of the annual variations in key hydroclimatic variables for both the reference (1981–2010) and future (2070–2099) periods, using the conventional and asynchronous methods for the Matane catchment. The figure mirrors the approach taken in Fig. 5 but focuses specifically on Matane, with the shaded areas indicating the variability across different climate models.

The trends observed in the Matane catchment largely reflect the broader findings across all catchments. For variables such as

interflow, streamflow, ETa, surface runoff, and SWE, the asynchronous and conventional methods exhibit patterns that align with the general trends noted in the overall analysis. However, the asynchronous method predicts significantly more groundwater recharge in the Matane catchment compared to the conventional method.

A key observation from Fig. 7 is the pronounced variability between climate models when using the asynchronous method. This variability suggests that some climate models may produce an unrealistic representation of physical processes,

particularly in relation to snow dynamics. For example, when examining the maximum snow water equivalent, the asynchronous method shows that the snowmelt period can start as early as April and extend as late as July, depending on the climate model. This is problematic, as it is highly unusual to have snow persist into July in the Matane region, making it unrealistic for a 30-year average to show such late snowmelt. This discrepancy raises concerns about the asynchronous method's ability to accurately simulate snow processes, a critical component of the hydrological cycle in regions with

significant winter snowfall.



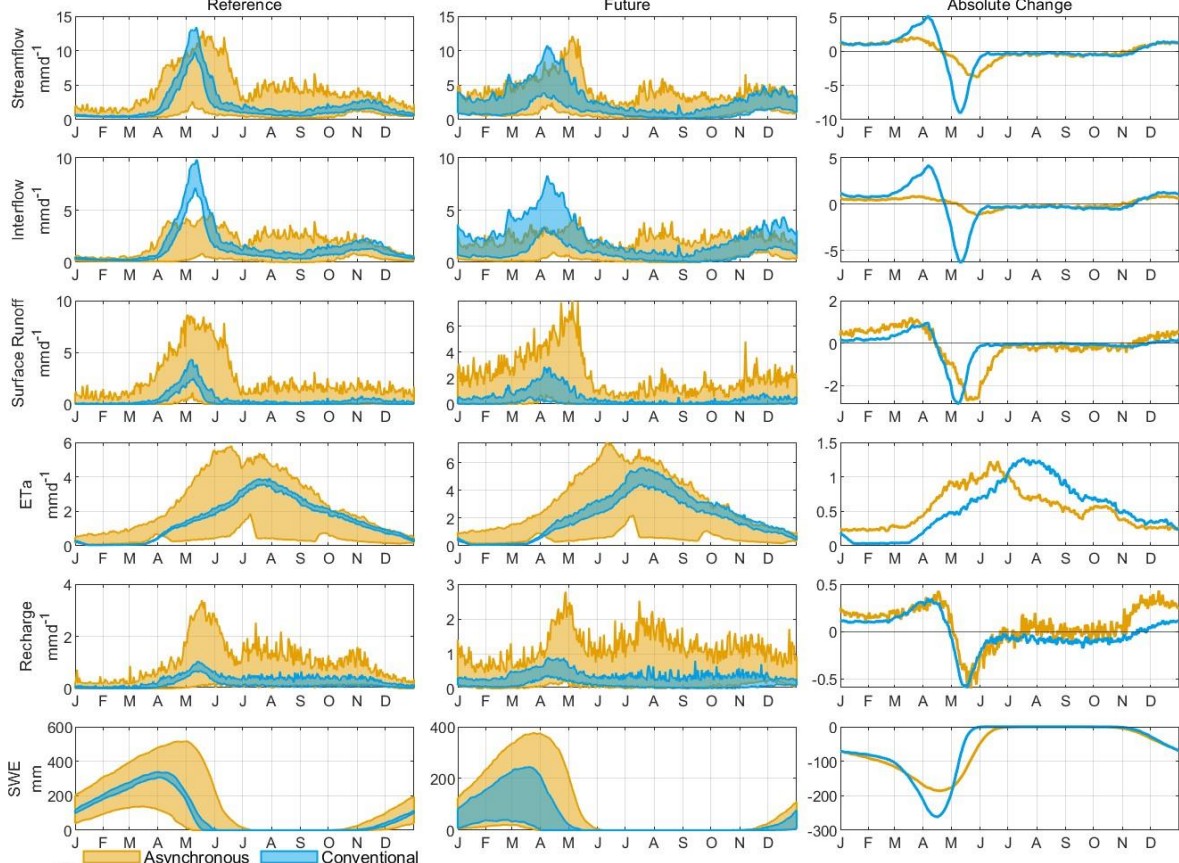

**Figure 7. Seasonal distribution of key hydroclimatic variables for the reference period (1981–2010), future period (2070–2099), and their average absolute changes across the Matane catchment using both the conventional and asynchronous methods. The figure presents the monthly averages of streamflow, interflow, surface runoff, actual evapotranspiration (ETa), groundwater recharge, and snow water equivalent (SWE). The left column shows the reference period, the middle column displays the future period, and the right column illustrates the absolute changes between the two periods. The shaded areas represent the variability across the climate models.**

Figure 8 presents a comparison of SWE for the reference period (1981–2010) across various climate models in the Matane Catchment. Each panel corresponds to a different climate model, with the shaded areas illustrating the range of annual variability. Similar figures for other catchments are provided in Appendix C (Fig. C1 to Fig. C9).

The figure clearly demonstrates the significant variability in SWE results produced by the asynchronous method compared to the conventional method. This disparity underscores the asynchronous method's challenges in accurately simulating snow patterns. For instance, the asynchronous method exhibits a broad range of SWE values, with the NorESM2-MM model showing only 200 mm of snow cover, while the IPSL-CM6A-LR model projects up to 700 mm for the same period. Additionally, the duration of snow accumulation varies widely within the asynchronous method. NorESM2-MM, for example, depicts snow presence from November to May, whereas EC-Earth3 suggests snow from October extending into July. These inconsistencies indicate a potential weakness in the asynchronous method's ability to capture snow dynamics.



In contrast, the conventional method consistently produces more stable and realistic results across different climate models, accurately reflecting the expected behaviour of snow accumulation and melt processes.

The poor representation of snow processes for asynchronous method is not limited to the Matane catchment. Other catchments, such as Valin and Godbout, exhibit similar anomalies, with snow present from November to September in some cases. Particularly striking is one climate model simulation in the Godbout catchment, where the asynchronous method predicts snow cover persisting throughout 11 months of the year. In the Bras du Nord catchment, the asynchronous method predicts roughly half the amount of snow compared to the conventional method, further highlighting again inadequate

representation of snow accumulation and melt dynamics.



**Figure 8. Snow water equivalent (SWE) comparison for the reference period (1981–2010) across various climate models for the Matane Catchment. This figure presents the SWE simulation results from multiple climate models using both the conventional**

**(blue) and asynchronous (yellow) methods. Each panel represents a different climate model, illustrating the seasonal SWE**





**accumulation and melt cycle. The shaded areas depict the range of annual variability, highlighting the spread of model outputs and the differences in snow dynamics as captured by each method.**

Figure 9 illustrates the spatial distribution of annual groundwater recharge rates in the Matane catchment for both the reference (1981–2010) and future (2070–2099) periods, comparing the results from the conventional and asynchronous

methods. Both methods exhibit similar spatial patterns, with higher elevations showing reduced recharge rates and lower elevations demonstrating higher recharge rates. An elevation map of the Matane catchment is provided in the Appendix D (Fig. D1). The asynchronous method, however, predicts a generally higher magnitude of recharge across the catchment.

When examining the absolute difference between the future and reference periods, both methods project a similar spatial pattern of changes in groundwater recharge, with a noticeable decrease in recharge at lower elevations. However, the

asynchronous method projects smaller increases in recharge in certain higher elevation areas, while the conventional method predicts a much more pronounced reduction—3 to 4 times greater—at lower elevations.

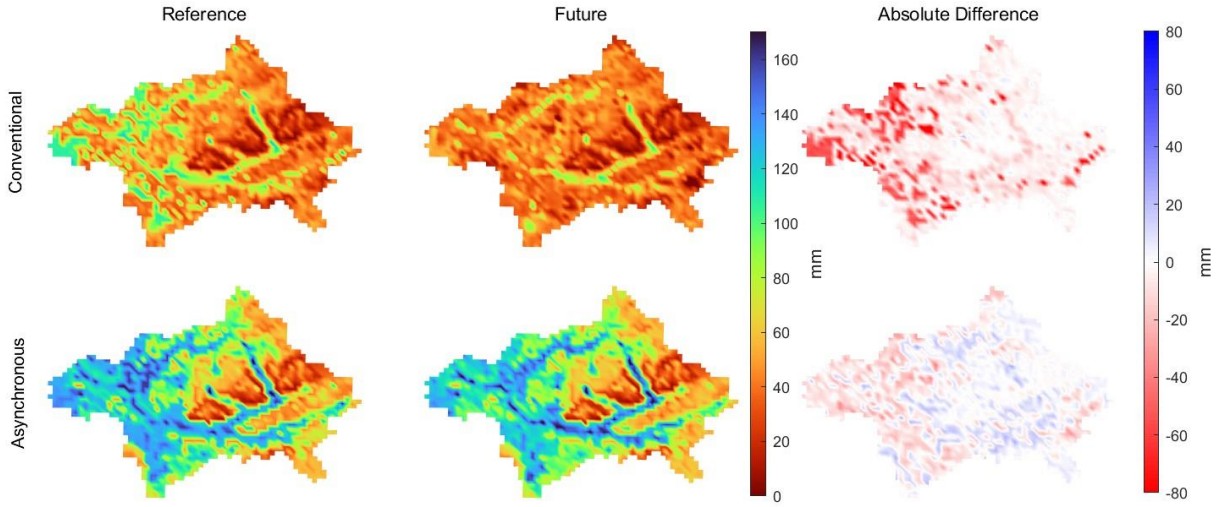

**Figure 9. Spatial distribution of annual groundwater recharge in the Matane catchment for the reference period (1981–2010) and future period (2070–2099) using both the conventional and asynchronous methods. The left column shows groundwater recharge**

**(mm) for the reference period, the middle column for the future period, and the right column illustrates the absolute difference between the two periods. The top row represents the conventional method, and the bottom row represents the asynchronous method. The color scale indicates groundwater recharge rates, with warmer colors representing lower recharge and cooler colors indicating higher recharge rates. The absolute difference maps highlight areas of significant change between the two periods, illustrating the spatial variability in projected groundwater recharge changes within the catchment.**

Figure 10 illustrates the spatial distribution of soil moisture across the Matane catchment for both the reference period (1981–2010) and the future period (2070–2099), comparing results from the conventional and asynchronous methods. Both methods demonstrate that soil moisture distribution is heavily influenced by soil type, as indicated by the consistent spatial patterns observed (detailed soil type information is provided in the Appendix D, Fig. D1). The soil moisture maps show that areas with finer soils, such as loam, tend to have higher moisture retention, while coarser soils, such as sandy loams, exhibit

lower moisture levels.





The asynchronous method tends to generate slightly higher soil moisture values compared to the conventional method, particularly in areas with inherently higher moisture retention capacity. The asynchronous method also displays greater variability in soil moisture patterns.

In terms of absolute changes between the reference and future periods, the conventional method projects a general decrease 540 in soil moisture, predominantly in regions with initially higher moisture values. The asynchronous method, while also projecting a decrease in soil moisture around high moisture areas, shows a more complex pattern with small regions exhibiting increases in soil moisture. Finally, it is noteworthy that the patterns of groundwater and soil moisture during the reference and future periods are spatially consistent and exhibit similar trends.

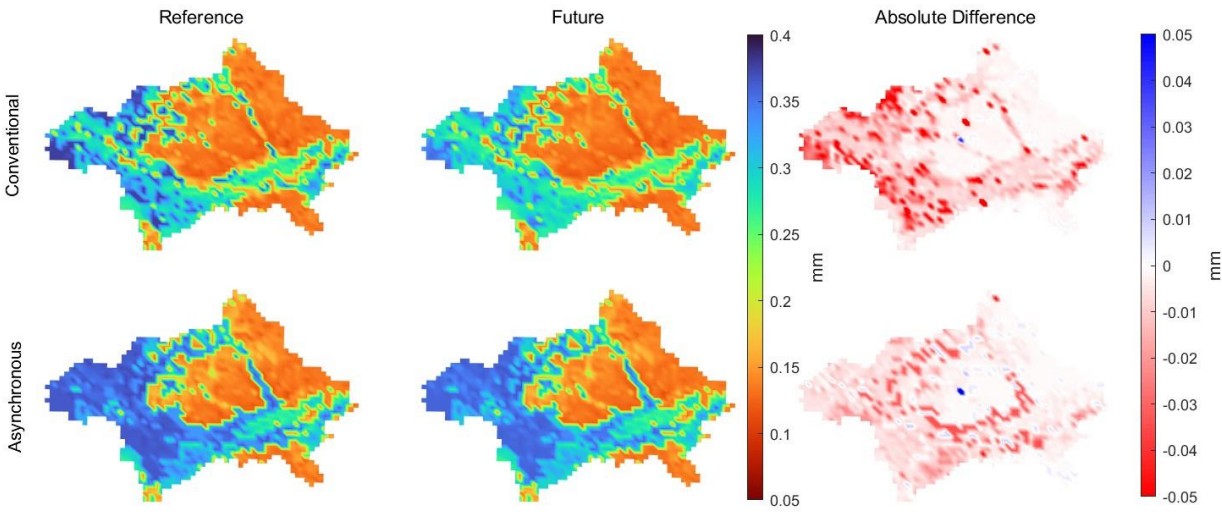

**Figure 10. Spatial distribution of soil moisture in the Matane catchment for the reference period (1981–2010) and future period (2070–2099) using both the conventional and asynchronous methods. The left column shows soil moisture for the reference period, the middle column for the future period, and the right column illustrates the absolute difference between the two periods. The top row represents the conventional method, and the bottom row represents the asynchronous method. The color scale indicates soil moisture levels, with warmer colors representing lower moisture content and cooler colors indicating higher moisture content. The** 550 **absolute difference maps highlight areas of significant change between the two periods, illustrating the spatial variability in projected soil moisture changes within the catchment.**

## 4 Discussion

The primary goal of this study was to evaluate the ability of the asynchronous method in reproducing key hydroclimatic processes within a catchment and to compare its performance according to the conventional method. Through detailed 555 analyses across multiple catchments, including a focused case study on the Matane catchment, this study aimed to determine whether the asynchronous method could offer a robust alternative to the conventional method, particularly in the context of climate change impact studies.



## 4.1 Hydroclimatic variables representation

One benefit of the asynchronous method is its ability to better represent extreme higher values (Q95 and Q90) compared to
conventional approach. By aligning the flow distribution with observed data, the asynchronous method effectively
reproduces the magnitude of these high flow events, which is critical for managing flood risks under future climate
conditions.

However, despite its strengths in representing streamflow distributions, this study's findings align with other research
indicating that the asynchronous method struggles to accurately capture the timing of observed streamflow, particularly
during spring high-flow events (Ricard et al., 2023). This issue contrasts with earlier findings by Ricard *et al.* (2020), who
reported that the asynchronous modeling approach provided a superior representation of the hydrologic regime compared to
the conventional method.

The asynchronous method performs comparably to the conventional method when it comes to representing the spatial
distribution of hydroclimatic variables such as soil moisture and groundwater recharge. This similarity is likely due to the
strong correlation between these hydroclimatic variables and the physical properties of the catchment, such as elevation and
soil type (Talbot et *al.*, 2024b). The consistent representation of these variables across both methods suggests that the
fundamental physical processes driving these patterns are well-captured, irrespective of the methodological differences in
how streamflow is simulated.

The asynchronous method shows consistency with the conventional method in predicting the overall trends for several
variables, such as increases in precipitation and ETa, as well as decreases in SWE and snowfall. These findings align with
broader climate change projections for the region, which anticipate warmer temperatures leading to reduced snow
accumulation and altered precipitation patterns (Aygün *et al.*, 2022; Nolin *et al.*, 2023; Valencia Giraldo *et al.*, 2023; Talbot
*et al.*, 2024b). However, while the overall trends may appear consistent, the underlying processes and the accuracy of the
projections differ significantly between the two methods.

Despite these consistencies, a significant issue with the asynchronous method is its high sensitivity to variability in climate
models. This problem comes from the biases inherent in climate models, which often lead to the simulation of hydrological
processes occurring either too early or too late (Chen et al., 2021; Ricard et al., 2023). The asynchronous method, as
currently implemented, adjust the calibration parameters to correct for biases, assuming that these biases remain constant
over time. Consequently, if a climate model has a significant or nonstationary biases, the asynchronous method will
perpetuate these biases, leading to inaccuracies in the timing of peak flows and the representation of hydroclimatic variables.
Ricard *et al.* (2023) also emphasizes this vulnerability, noting that the asynchronous method is particularly prone to
producing outlying projections due to the uncorrected biases in raw climate model outputs.

One of the most critical issues highlighted in this study is the concept of equifinality, where different models or methods
achieve similar outcomes for different reasons (Mei et al., 2023; Yassin et al., 2017). In the case of the asynchronous



method, it appears to replicate certain aspects of the conventional method's projections, but it does so through potentially flawed mechanisms.

The asynchronous method struggles to synchronize streamflow with the actual timing of hydrological events, particularly snowmelt. This lack of synchronization leads to a cascade of flawed mechanisms throughout the model. For instance, when snowmelt occurs too early or too late, the timing and magnitude of surface runoff are inaccurately represented, which can

lead to unrealistic increases in surface runoff during inappropriate seasons. This misalignment also affects evapotranspiration and groundwater recharge, causing large, unrealistic variations, further skewing the model's output. Because the streamflow is not properly synchronized with the seasonal dynamics, the asynchronous method ultimately produces streamflow simulations that may match the overall distribution of the observations but do so for the wrong reasons.

Equifinality becomes particularly problematic in this context because the asynchronous method may achieve similar

projected changes in hydroclimatic variables as the conventional method, but for reasons that are not hydrologically sound. This brings into question the reliability of its projections, especially when the method demonstrates high variability among different climate models. Such variability, coupled with the method's inability to accurately replicate key hydrological processes, suggests that the asynchronous method, as implemented in this study, may not provide a robust framework for analyzing climate change impacts on hydroclimatic variables.

In contrast, the conventional method excels at addressing the timing of hydrological events due to its optimization process, which incorporates the synchronization between observed and simulated flows. This synchronization allows the model to more accurately capture the timing of hydrological processes, such as snowmelt and peak flows. Given the importance of this synchronicity, it would be beneficial for the asynchronous method to integrate a similar measure of event timing. This adjustment could pave the way for a semi-asynchronous approach that balances the strengths of both methods, offering better

overall performance in hydrological simulations.

## 4.2 Advantages and limitations

Both the conventional and asynchronous methods present distinct advantages and limitations. One of the most notable distinctions between the two methods lies in how they handle extremes. The Multivariate Bias Correction (MBCn) approach, typically used in the conventional method, tends to dampen the extremes, smoothing out the peaks. In contrast, the

asynchronous method, which calibrates directly on the distribution of streamflow without bias correction, preserves (or attempts to preserve) these extremes (Ricard et al., 2023). Maintaining extreme values may provide a more realistic representation of potential high-impact events.

Another critical consideration is the computational demand of the asynchronous method. Due to its reliance calibration on every climate model, the asynchronous method requires significantly more computational time in the calibration process.

This increased computational cost must be weighed against the benefits of using the asynchronous method, particularly when the conventional method might achieve similar results with less computational effort and more established reliability.





The performance of the asynchronous method in snow-dominated catchments has proven to be problematic in this study. The method's inability to accurately capture snowmelt processes, as evidenced by the unrealistic snow retention and melt timing, casts doubt on its utility in regions where snow dynamics play a critical role in the hydrological cycle. Additionally, the high variability observed between climate models when using the asynchronous method suggests that the approach may be overly sensitive to the inherent uncertainties present in raw climate data. This variability complicates the interpretation of results and diminishes confidence in the method's projections, particularly in scenarios where precise predictions are required for decision-making.

The key takeaway from this study is that while the asynchronous method allowed preserving the distribution of streamflow and maintaining extremes, it does so at the cost of increased variability and potential inaccuracies in simulating critical hydrological processes, particularly those related to snow. Therefore, the asynchronous method, as implemented in this study, should be used with caution, especially in snow-dominated catchments where accurate representation of snowmelt is crucial. However, the asynchronous method could be useful in scenarios where the distribution of extremes is of particular interest, such as in regions where the temporal distribution of streamflow is less critical than the overall volume

The conventional method, which is optimize to account for event-specific dynamics, remains the more reliable option for most applications, particularly when the goal is to simulate the timing and magnitude of streamflow events with a higher degree of accuracy.

Ultimately, these findings aim to inform decision-making in critical sectors such as agriculture, water resource management, urban planning, and environmental conservation. For example, soil moisture data is essential in agriculture for optimizing irrigation and improving crop yields, as well as in environmental management for maintaining wetlands and forest ecosystems. Groundwater recharge data supports sustainable management of aquifers, which is crucial for drinking water supplies, agriculture, and industrial use, while also guiding urban planning to avoid flooding or subsidence. Surface runoff modeling is vital for flood prevention and urban infrastructure design, ensuring stormwater systems can handle heavy rainfall. Lastly, streamflow data is key to water resource management, enabling efficient allocation for agriculture and industry, flood forecasting, and optimizing hydroelectric power generation. By providing detailed projections of these key hydroclimatic variables, this study supports adaptive management strategies across a wide range of sectors impacted by climate change.

## 4.3 Future directions

Looking forward, one of the most promising avenues for improving the asynchronous method is the integration of synchronicity, leading to the development of a semi-asynchronous approach. This hybrid method would combine the strengths of both the conventional and asynchronous methods, offering a more balanced solution that mitigates the weaknesses observed in each. By incorporating synchronicity into the calibration process, the semi-asynchronous method would better align the timing of hydrological events, such as snowmelt, with observed data, improving its ability to capture critical seasonal dynamics.





655 For instance, modifying the objective function to calibrate based on seasonal or monthly data could enhance the model's ability to simulate hydrological processes. This integration of event timing into the calibration process is crucial for addressing the timing discrepancies that currently limit the asynchronous method's performance.

In parallel, ongoing advancements in climate modeling provide an opportunity to further refine the semi-asynchronous approach. As climate models become more accurate, with fewer biases and enhanced temporal precision, the challenges of 660 synchronization in the current asynchronous method could be significantly alleviated. These improvements would enable the semi-asynchronous method to offer more robust and reliable simulations of hydrological processes under future climate scenarios, positioning it as a more versatile tool for climate impact assessments.

## 5 Conclusion

This study aimed to evaluate the performance of the asynchronous method in comparison to the conventional method for 665 simulating key hydroclimatic variables within catchments, with a focus on the implications for climate change impact studies. Through a detailed analysis of multiple catchments, including a focused case study on the Matane catchment, the study has revealed several important insights into the strengths and limitations of both methods.

The findings indicate that while the asynchronous method shows promise in accurately preserving extreme values, it struggles significantly with the timing of hydrological events, particularly those related to snowmelt. This timing issue is 670 critical in snow-dominated catchments, where accurate snowmelt representation is crucial for reliable hydrological modeling. The asynchronous method's vulnerability to equifinality, nonstationarity and biases in climate models further complicates its application, often leading to increased variability and potential inaccuracies in key hydrological processes which may not be hydrologically sound.

In contrast, the conventional method, with its bias correction step, provides more reliable simulations of event-specific 675 dynamics, particularly in capturing the timing and magnitude of streamflow events, but at cost of underestimating extreme hydrological events. This reliability makes it a more suitable choice for most hydrological applications, especially in regions where precise timing of hydrological events is essential.

From a practical standpoint, while the asynchronous method offers the advantage of preserving extremes, it comes at the cost of increased computational demand and variability in projections, which may limit its utility in certain contexts. The 680 conventional method, on the other hand, remains a robust and reliable tool for simulating hydrological processes under future climate scenarios, particularly when accuracy in timing is a critical factor.

Looking ahead, there are significant opportunities to refine the asynchronous method, particularly by integrating synchronicity into the calibration process to better capture seasonal dynamics. This enhancement could lead to the development of a semi-asynchronous approach that combines the strengths of both the asynchronous and conventional 685 methods, addressing the current challenges related to event timing while preserving the ability to model extremes. Such a





hybrid method would offer a more balanced solution, improving accuracy in snowmelt representation and other critical hydrological processes.

Until these refinements are realized, the asynchronous method should be applied with caution, especially in regions where precise seasonal dynamics, such as snowmelt, are critical. Future research should focus on advancing the semi-asynchronous method, ultimately aiming to create a more versatile and robust tool for hydrological modeling in the context of climate change.


**Appendix A**

**Table A1. Kling-Gupta Efficiency (KGE) values for the conventional method during the calibration and validation periods across ten catchments. The mean KGE values for calibration and validation are also provided.**

| Conventional Method | | KGE | |
|---|---|---|---|
| Name | Area (km$^2$) | Calibration | Validation |
| Bonaventure | 1910 | 0.847 | 0.889 |
| Matane | 1650 | 0.906 | 0.906 |
| Ouelle | 795 | 0.894 | 0.834 |
| Bécancour | 919 | 0.850 | 0.807 |
| Nicolet Sud-Ouest | 549 | 0.817 | 0.786 |
| Au Saumon | 738 | 0.831 | 0.778 |
| Bras du Nord | 642 | 0.873 | 0.872 |
| Du Loup | 774 | 0.838 | 0.804 |
| Valin | 746 | 0.902 | 0.885 |
| Godbout | 1570 | 0.869 | 0.863 |
| | Mean | 0.863 | 0.842 |





**Table A2. Root Mean Square Error (RMSE) values for the asynchronous method during the calibration and validation periods across ten catchments. The mean RMSE values for calibration and validation are also provided.**

| Asynchronous Method | | RMSE | | | |
|---|---|---|---|---|---|
| Name | Area (km$^2$) | Calibration | | Validation | |
| | | Mean | Std | Mean | Std |
| Bonaventure | 1910 | 0.108 | 0.031 | 0.154 | 0.047 |
| Matane | 1650 | 0.124 | 0.021 | 0.180 | 0.034 |
| Ouelle | 795 | 0.125 | 0.027 | 0.167 | 0.043 |
| Bécancour | 919 | 0.095 | 0.019 | 0.158 | 0.048 |
| Nicolet Sud-Ouest | 549 | 0.125 | 0.034 | 0.200 | 0.097 |
| Au Saumon | 738 | 0.156 | 0.054 | 0.209 | 0.053 |
| Bras du Nord | 642 | 0.175 | 0.044 | 0.243 | 0.063 |
| Du Loup | 774 | 0.085 | 0.019 | 0.165 | 0.091 |
| Valin | 746 | 0.109 | 0.036 | 0.162 | 0.055 |
| Godbout | 1570 | 0.110 | 0.032 | 0.157 | 0.037 |
| | Mean | 0.121 | 0.031 | 0.179 | 0.057 |





## Appendix B

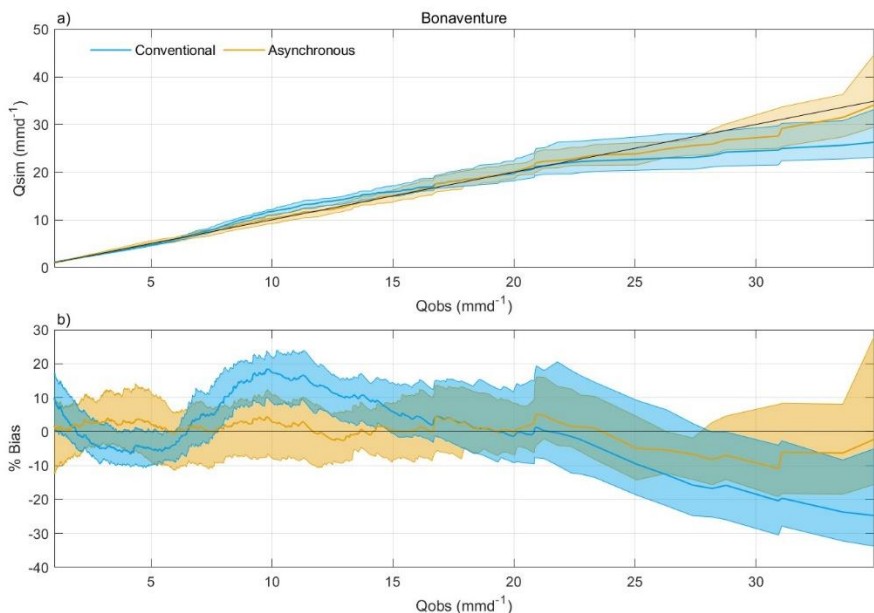


**Figure B1. Performance comparison between the conventional and asynchronous methods for the Bonaventure catchment during the reference period (1981–2010). Panel (a) displays the relationship between simulated daily streamflow (Qsim) and observed daily streamflow (Qobs) for the conventional (blue) and asynchronous (yellow) methods. The x-axis represents the observed daily streamflow, while the y-axis represents the simulated streamflow. Panel (b) shows the percentage bias between observed and**
**simulated streamflows for both methods, with the x-axis representing the observed daily streamflow and the y-axis displaying the percentage bias relative to the observed values. The shaded regions in both panels illustrate the variability among climate models around the mean bias for each method, emphasizing the differences in how well each method simulates streamflow across the observed streamflow range.**




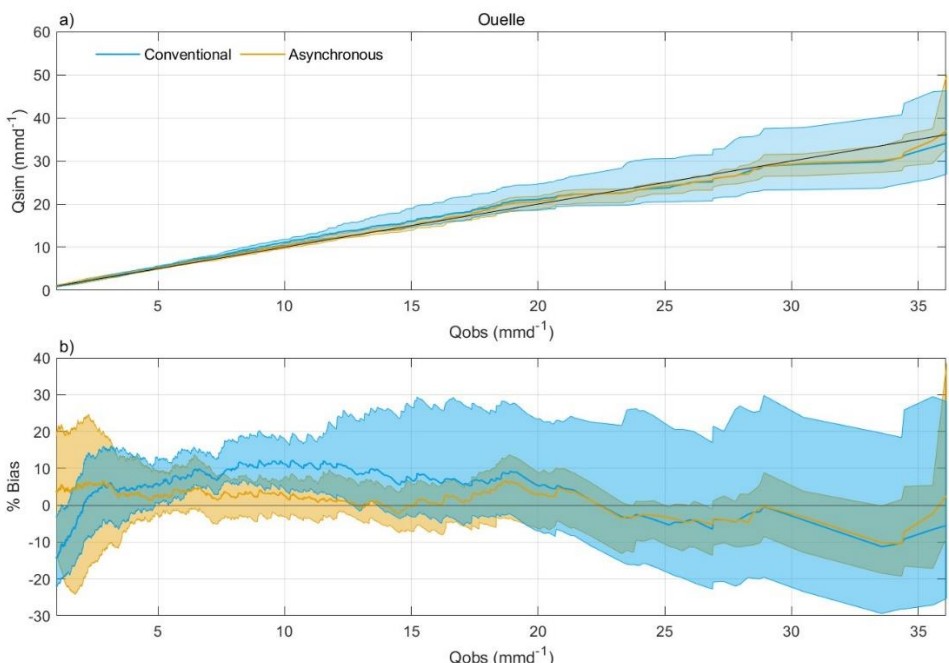

**Figure B2. Same as Fig. B1, but for Ouelle catchment.**

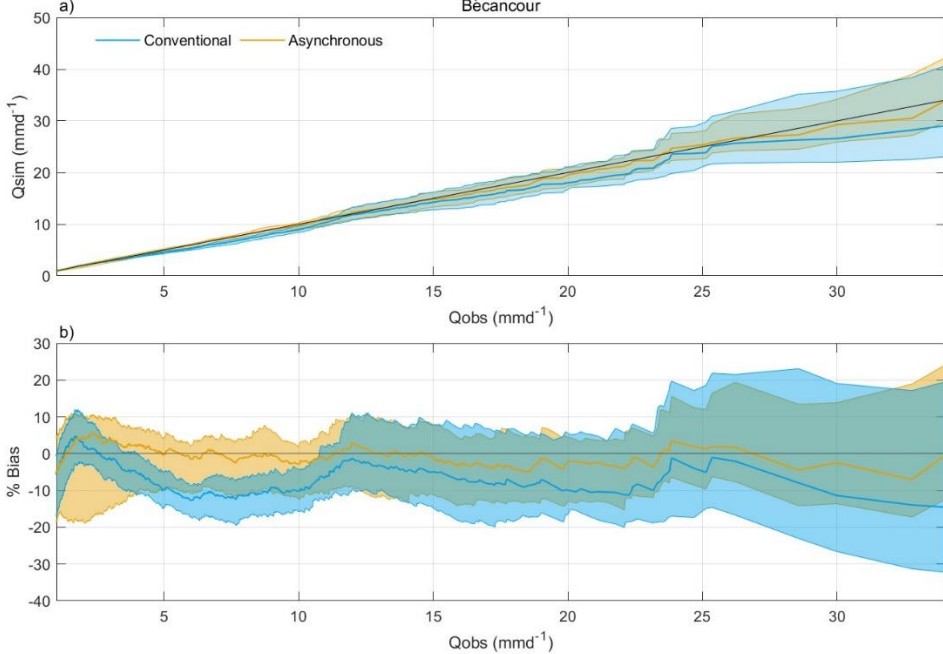

**Figure B3. Same as Fig. B1, but for Bécancour catchment.**





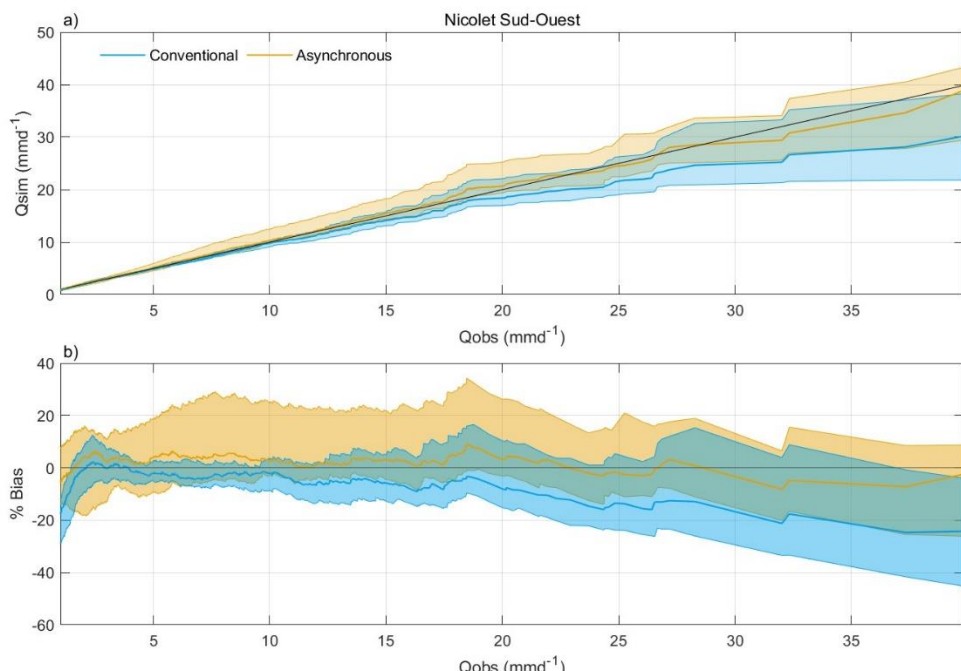

**Figure B4. Same as Fig. B1, but for Nicolet Sud-Ouest catchment.**

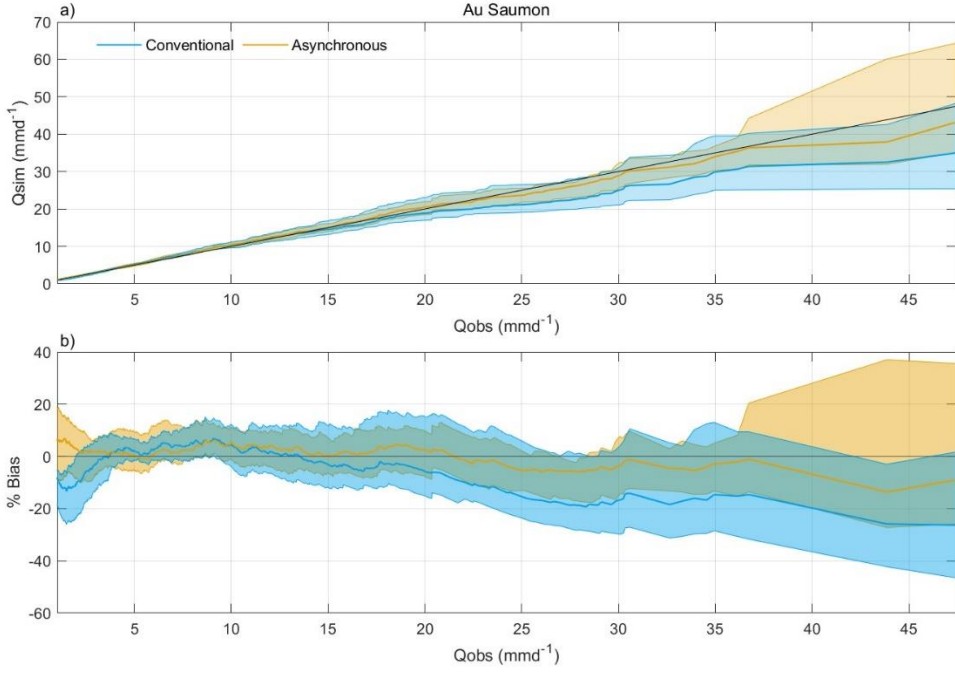


**Figure B5. Same as Fig. B1, but for Au Saumon catchment.**



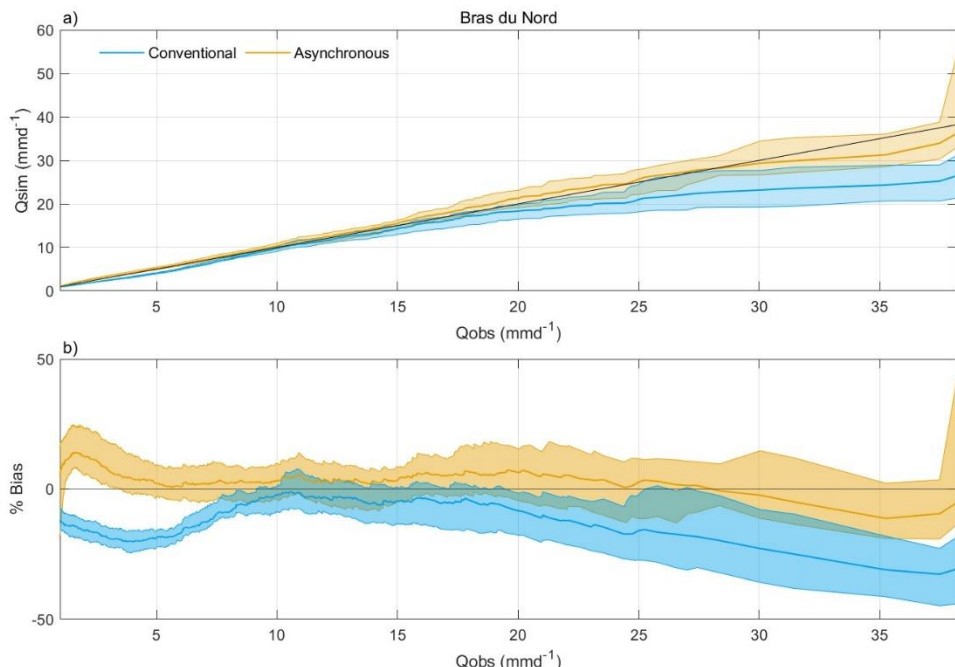

**Figure B6. Same as Fig. B1, but for Bras du Nord catchment.**

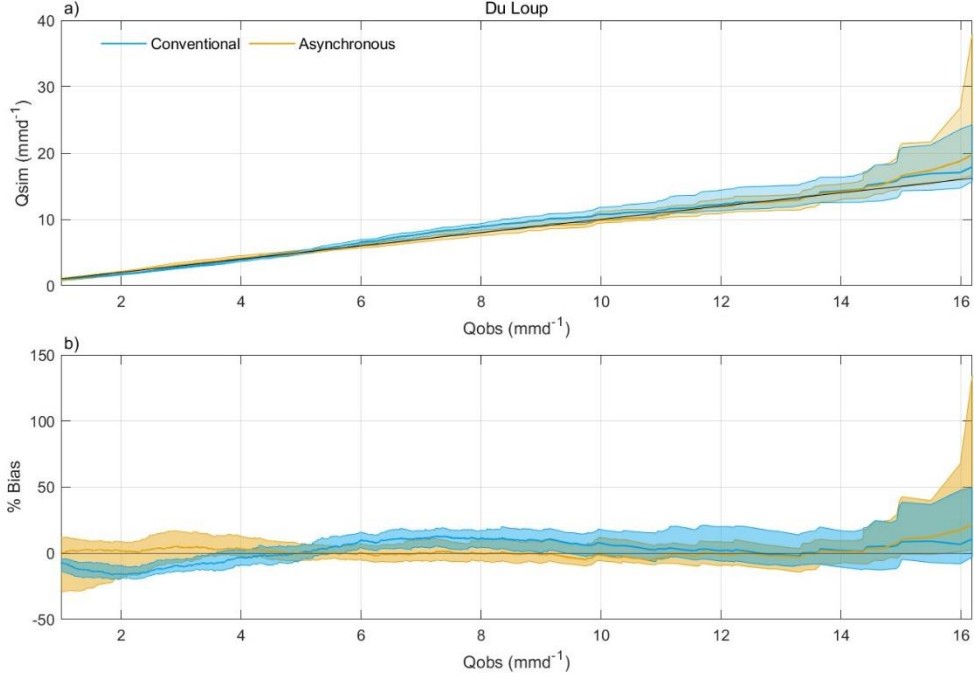

**Figure B7. Same as Fig. B1, but for Du Loup catchment.**





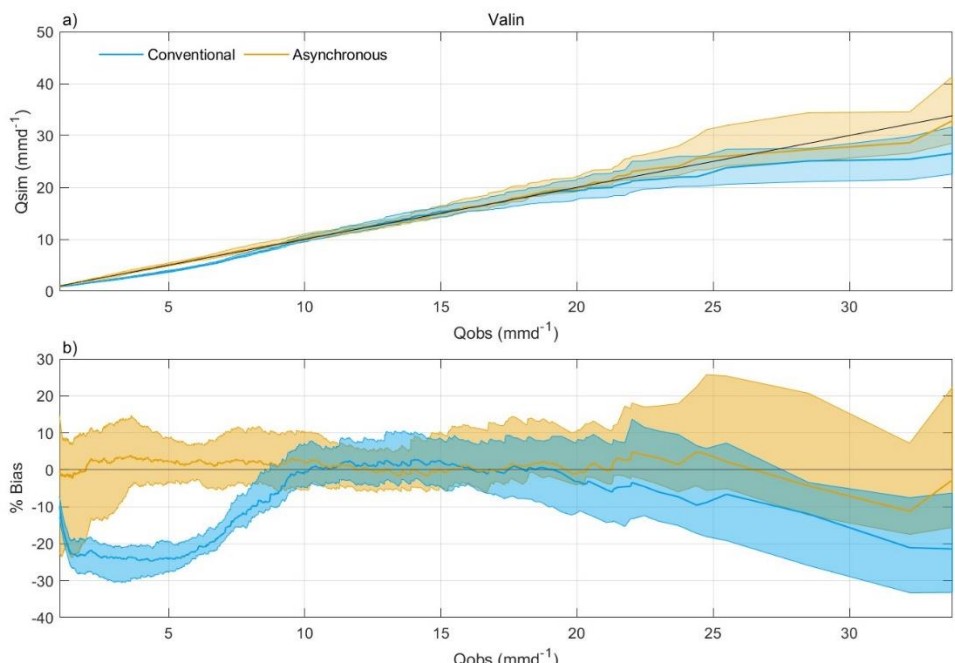

**Figure B8. Same as Fig. B1, but for Valin catchment.**

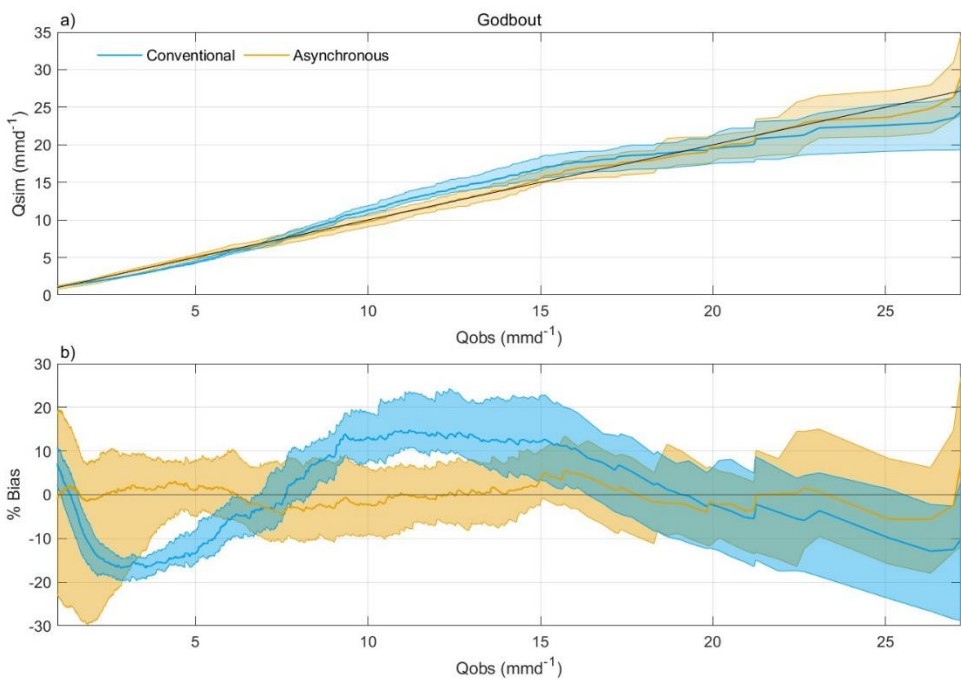

**Figure B9. Same as Fig. B1, but for Godbout catchment.**




## Appendix C

**Figure C1. Snow water equivalent (SWE) comparison for the reference period (1981–2010) across various climate models for the Bonaventure Catchment. This figure presents the SWE simulation results from multiple climate models using both the conventional (blue) and asynchronous (yellow) methods. Each panel represents a different climate model, illustrating the seasonal SWE accumulation and melt cycle. The shaded areas depict the range of annual variability, highlighting the spread of model outputs and the differences in snow dynamics as captured by each method.**





**Figure C2. Same as Fig. C1, but for Ouelle catchment.**






**Figure C3. Same as Fig. C1, but for Bécancour catchment.**







**Figure C4. Same as Fig. C1, but for Nicolet Sud-Ouest catchment.**





Figure C5. Same as Fig. C1, but for Au Saumon catchment.



**Figure C6. Same as Fig. C1, but for Bras du Nord catchment.**



**Figure C7. Same as Fig. C1, but for Du Loup catchment.**





**Figure C8. Same as Fig. C1, but for Valin catchment.**






**Figure C9. Same as Fig. C1, but for Godbout catchment.**





**Appendix D**

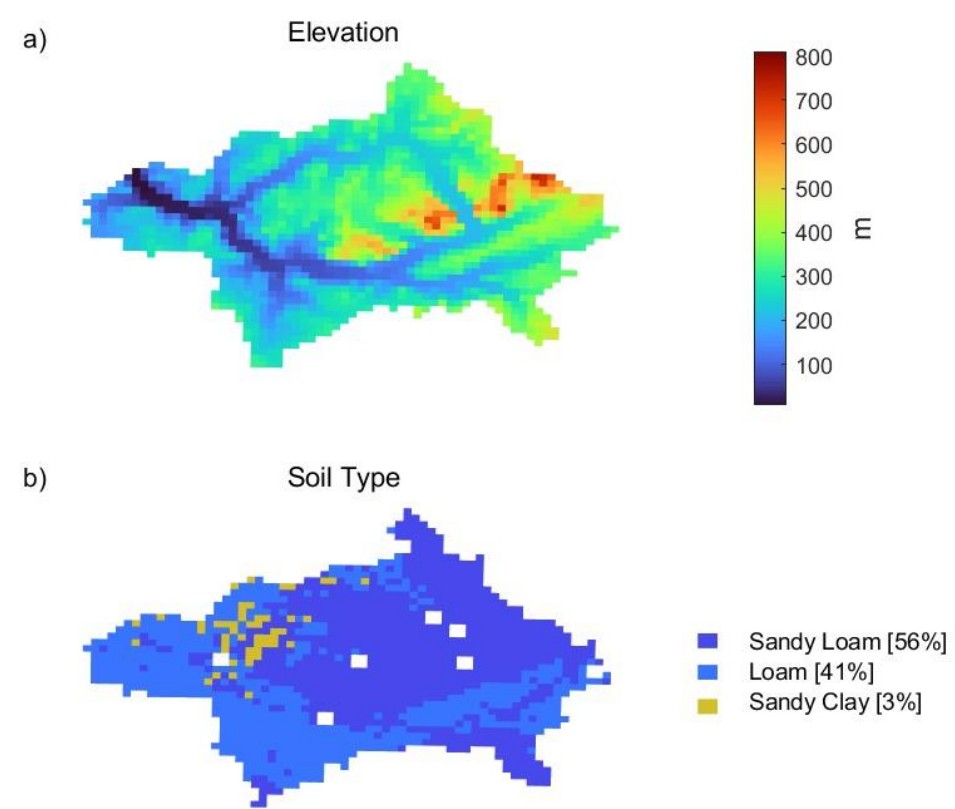


**Figure D1. Topographic and soil type characteristics of the Matane catchment. Panel (a) shows the elevation map, with elevations ranging from 100 to 800 meters above sea level. Higher elevations are indicated in warmer colors (reds and oranges), while lower elevations are shown in cooler colors (blues and greens). Panel (b) displays the distribution of soil types within the catchment, with sandy loam covering 56% of the area (dark blue), loam covering 41% (light blue), and sandy clay occupying 3% (yellow).**

**Code and data availability**

The calibrated WaSiM model and all simulations for all catchments discussed in this study is publicly accessible at https://osf.io/n87ey/ (Talbot et al., 2024c).

**Author contribution**

FT, JDS, SR, and RA contributed to the conceptualization and methodology design of the study. FT performed the formal

analysis, investigation, data curation, and conducted the model simulations. JLM contributed to the MBCn post-processing of climate data ensembles. FT was responsible for visualization and led the original draft preparation. JDS and RA provided





supervision and project administration. AP, GD, JDS, RA, SR, JLM and FT contributed to the writing, review, and editing of the manuscript.

**Competing interests**

The authors declare that they have no conflict of interest.

**Acknowledgements**

This work was funded jointly by the ministère des Ressources naturelles et des Forêts (Quebec, Canada, project number 112332187 conducted at the Direction de la recherche forestière and led by Jean-Daniel Sylvain) and the Forest research service contract number 3322-2022-2187-01 obtained by Richard Arsenault from the Ministère des Ressources naturelles et

des Forêts (Quebec, Canada). The authors also acknowledge the use of ChatGPT-4 for assistance in correcting spelling mistakes and improving the flow of text during the manuscript preparation process. The base map in Fig. 1 was created using ArcGIS® software by Esri. ArcGIS® and ArcMap™ are the intellectual property of Esri and are used herein under license. Copyright © Esri. All rights reserved. For more information about Esri® software, please visit www.esri.com.



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
