# Peer review of "Towards a semi-asynchronous method for hydrological modeling in climate change studies"

_EGUsphere, 2024_

## Author Comment (AC1)

**Towards a semi-asynchronous method for hydrological modeling in climate change studies**

Frédéric Talbot[1], Simon Ricard[3], Jean-Daniel Sylvain[2], Guillaume Drolet[2], Annie Poulin[1], Jean-Luc Martel[1], Richard Arsenault[1]

[1] Hydrology, Climate and Climate Change Laboratory, École de technologie supérieure, Université du Québec, Montréal, H3C 1K3, Canada

[2] Direction de la recherche forestière, Ministère des Ressources naturelles et des Forêts, Québec, G1P 3W8, Canada

[3] Sciences and Engineering, Université Laval, Québec, G1V 0A6, Canada

*Correspondence to*: Frédéric Talbot (frederic.talbot.2@ens.etsmtl.ca)

**RC1 (https://doi.org/10.5194/egusphere-2024-3037-RC1)**

The manuscript of Talbot et al. provides a comparison of two approaches for projecting climate change impacts on the terrestrial hydrological cycle at the catchment level, namely the conventional and the asynchronous approaches. The two approaches are accurately applied over 10 small-to-medium extent catchments in southern Quebec, which are particularly affected by snow dynamics (accumulation and melting). Basically, the difference between the two approaches consists of correcting the biases of the historical climate simulations either directly (i.e., applying a bias correction method) or through a specific calibration of the parameters of the hydrological model (i.e., the calibrated parameters "incorporate" the climatological bias). This second approach is risky since, as correctly discussed by the authors, it can easily lead to flawed mechanisms. And, to be honest, it doesn't convince me much, primarily for this drawback but also because it requires a more considerable computational effort. Nevertheless, I agree that it should be thoroughly tested and verified.

Though the paper is potentially engaging and thought-provoking, I have several comments that should be addressed before publication.

We sincerely appreciate the reviewer acknowledgment that our work is accurate and that the paper is engaging and thought-provoking, which reinforces the value of our research and the soundness of our hypothesis. The reviewer's thorough and constructive feedback has significantly contributed to enhancing the clarity, robustness, and overall impact of our manuscript. Below, we address each of the concerns and outline the corresponding revisions.

First, as many studies before demonstrated, most of the reliability of the climate projection depends on GCMs, which are the primary sources of uncertainty (e.g., 10.1016/j.jhydrol.2012.11.062, 10.1007/s00382-019-04664-w, 10.1016/j.ejrh.2022.101120, 10.1002/joc.8661 and several others). In some cases, bias correction is mandatory to achieve meaningful hydrological output; in other cases, in which the historical simulations perform better, one can even think of avoiding any bias correction. So, the first point is showing a preliminary analysis of the performance of the GCMs, some of which could even be deleted if performing too badly (e.g., in my experience, some of them are neither able to reproduce precipitation seasonality correctly).

The reviewer raises an important point regarding the evaluation of GCM performance prior to their use in hydrological modeling. While Ricard et al. (2023) (https://doi.org/10.5194/hess-27-2375-2023) demonstrated that selecting GCMs based on hydrologic performance generally did not significantly alter conclusions compared to using the full ensemble, we acknowledge the value of assessing GCM performance to ensure robust model outputs, and ultimately enhancing the reliability of the results for the benefit of the readers.

In response, we propose to add two additional tables in Section 2.2.5 summarizing key performance indicators of the selected GCMs. These tables highlight the ability of each model to reproduce historical precipitation seasonality (Table 2) and temperature variability (Table 3), evaluated against ERA5 reanalysis data. These metrics provide a clearer picture of the GCMs' performance in representing key climatic features.

We will also clarify that all climate models were retained to maintain consistency with broader climate impact assessments.

**Table 2. Monthly precipitation bias (%) between 18 global climate models (GCMs) and ERA5 reanalysis data for the reference period (1981-2010). Values represent the average bias across the 10 catchments included in the study. Monthly biases are shown for January (J) through December (D), with the final column indicating the annual mean bias for each model.**

| | | | | | Monthly precipitation bias (%) | | | | | | | | |
|---|---|---|---|---|---|---|---|---|---|---|---|---|---|
| **GCM Name** | **J** | **F** | **M** | **A** | **M** | **J** | **J** | **A** | **S** | **O** | **N** | **D** | **Mean** |
| ACCESS-ESM1-5 | 14.1 | 22.1 | 29.6 | 2.5 | 13.7 | 12.6 | 23.4 | 11.2 | -16.2 | -16.3 | -1.8 | 8.2 | **8.6** |
| CMCC-ESM2 | -12.1 | 7.3 | -6.8 | -14.7 | -12.8 | -11.1 | -0.9 | -14.9 | -16.4 | -11.0 | -2.1 | -9.2 | **-8.7** |
| CanESM5 | -10.3 | 0.6 | -2.0 | -6.6 | 16.5 | 23.9 | 36.2 | 26.6 | 22.0 | 5.4 | -4.8 | -23.4 | **7.0** |
| EC-Earth3 | -1.3 | 26.1 | 26.7 | -2.6 | 1.6 | 8.3 | 0.7 | -6.9 | -7.5 | 3.6 | 3.0 | -2.4 | **4.1** |
| EC-Earth3-CC | -0.5 | 37.2 | 18.3 | 5.0 | 11.3 | 8.5 | 0.4 | -9.5 | 13.8 | 11.7 | 7.4 | -0.5 | **8.6** |
| EC-Earth3-Veg-LR | -2.0 | 10.6 | 11.1 | -11.3 | 4.6 | -0.4 | -3.6 | -12.4 | 1.1 | 6.0 | -2.1 | -4.3 | **-0.2** |
| FGOALS-g3 | -3.8 | -11.8 | -7.3 | -29.0 | -38.9 | -32.5 | -22.1 | -17.1 | -23.0 | -13.9 | -1.5 | 3.7 | **-16.4** |
| GFDL-ESM4 | -5.9 | 12.2 | 3.2 | -0.7 | 1.5 | 18.5 | 11.1 | 0.7 | 7.3 | 7.8 | -1.0 | -16.4 | **3.2** |
| INM-CM4-8 | 20.5 | 22.6 | 26.0 | -5.6 | 11.0 | 9.0 | 13.6 | 9.3 | -18.3 | -0.6 | 11.2 | 20.1 | **9.9** |
| INM-CM5-0 | 18.1 | 18.7 | 16.0 | -0.8 | 9.7 | 12.5 | 23.5 | 11.2 | -8.2 | -16.1 | 15.2 | 6.6 | **8.9** |
| IPSL-CM6A-LR | 27.2 | 39.6 | 31.4 | 0.3 | 8.4 | 16.2 | 34.5 | 28.4 | 10.7 | -1.7 | 3.1 | 12.1 | **17.5** |
| MIROC6 | 17.2 | 17.2 | 3.9 | -13.5 | 16.9 | 27.8 | 22.1 | 10.1 | -4.7 | 9.2 | 3.3 | -4.6 | **8.7** |
| MPI-ESM1-2-HR | -4.3 | 14.4 | 24.9 | 22.9 | 20.2 | 13.3 | 12.9 | 1.2 | 1.4 | 9.8 | 4.6 | -18.7 | **8.5** |
| MPI-ESM1-2-LR | -4.0 | 11.9 | 15.5 | -0.8 | 6.9 | 10.5 | 22.0 | -12.0 | 8.0 | -10.1 | -6.4 | -17.9 | **2.0** |
| MRI-ESM2-0 | -8.7 | 2.6 | -3.0 | -10.0 | -0.1 | 8.3 | 10.3 | -6.5 | 0.3 | 0.6 | -4.6 | 5.8 | **-0.4** |
| NESM3 | 0.7 | 38.6 | 36.1 | 10.1 | 8.0 | 7.9 | 17.9 | 5.5 | -9.5 | -8.9 | 1.2 | -13.4 | **7.8** |
| NorESM2-LM | -16.5 | 1.7 | -9.3 | -25.1 | -22.8 | -10.9 | -23.9 | -23.2 | -12.2 | -15.9 | -18.5 | -20.1 | **-16.4** |
| NorESM2-MM | -10.9 | -6.5 | -3.3 | -23.7 | -20.2 | -7.4 | -15.4 | -26.1 | -20.0 | -18.4 | -5.0 | -5.8 | **-13.5** |

50

**Table 3. Monthly temperature bias (°C) between 18 global climate models (GCMs) and ERA5 reanalysis data for the reference period (1981-2010). Values represent the average bias across the 10 catchments included in the study. Monthly biases are shown for January (J) through December (D), with the final column indicating the annual mean bias for each model.**

**Monthly temperature bias (°C)**

| GCM Name | J | F | M | A | M | J | J | A | S | O | N | D | Mean |
|---|---|---|---|---|---|---|---|---|---|---|---|---|---|
| ACCESS-ESM1-5 | 4.4 | 2.3 | 2.6 | 1.3 | -0.3 | 0.1 | 0.6 | 0.4 | 0.9 | 1.5 | 1.4 | 4.1 | **1.6** |
| CMCC-ESM2 | 0.0 | -0.2 | 0.6 | 0.5 | 0.0 | -0.3 | 0.3 | 0.2 | 0.7 | 1.2 | 0.7 | 0.3 | **0.3** |
| CanESM5 | 0.1 | -0.3 | 0.9 | 1.6 | 2.6 | 2.0 | 1.4 | 1.9 | 1.4 | 0.8 | -0.2 | -0.4 | **1.0** |
| EC-Earth3 | -2.2 | -1.7 | -2.2 | -3.5 | -4.6 | -3.0 | -2.2 | -1.0 | -0.2 | -0.7 | -0.9 | -1.5 | **-2.0** |
| EC-Earth3-CC | -1.3 | 0.2 | -1.5 | -2.3 | -3.1 | -2.1 | -1.2 | -0.4 | 0.4 | 0.6 | -1.0 | -1.2 | **-1.1** |
| EC-Earth3-Veg-LR | -3.2 | -4.1 | -2.7 | -3.5 | -4.3 | -3.4 | -1.4 | -0.6 | 0.5 | 0.0 | -1.6 | -2.6 | **-2.2** |
| FGOALS-g3 | -1.2 | -2.0 | -0.7 | 0.0 | -1.1 | -0.6 | 0.2 | 0.1 | -0.9 | -0.3 | -0.2 | -0.4 | **-0.6** |
| GFDL-ESM4 | -0.6 | 0.1 | -1.3 | -2.0 | -2.6 | -1.9 | -1.9 | -1.9 | -1.5 | -1.0 | -1.8 | -0.6 | **-1.4** |
| INM-CM4-8 | -1.2 | -0.7 | -1.2 | -0.3 | 1.5 | 0.8 | 0.7 | 0.6 | 0.7 | 1.8 | 1.8 | 0.5 | **0.4** |
| INM-CM5-0 | -2.4 | -3.1 | -3.6 | -2.9 | 0.3 | 0.5 | 0.5 | 0.1 | 0.1 | 1.8 | 1.4 | -0.8 | **-0.7** |
| IPSL-CM6A-LR | -2.0 | -1.5 | -1.7 | -1.9 | -0.4 | 1.0 | 2.0 | 2.7 | 1.4 | 0.2 | -1.2 | -1.8 | **-0.3** |
| MIROC6 | 2.5 | 1.1 | 0.0 | 0.1 | 1.9 | 2.5 | 2.7 | 2.3 | 2.4 | 2.3 | 1.7 | 1.7 | **1.8** |
| MPI-ESM1-2-HR | 0.3 | -1.1 | -1.8 | -1.0 | -0.1 | 0.0 | -0.4 | -0.4 | 0.0 | -0.2 | -0.6 | 0.2 | **-0.4** |
| MPI-ESM1-2-LR | -1.1 | -2.1 | -2.4 | -1.3 | 0.0 | -0.4 | -0.5 | -0.8 | -0.3 | -0.1 | -1.5 | -1.0 | **-1.0** |
| MRI-ESM2-0 | -0.5 | -0.4 | -0.2 | -0.4 | -0.8 | -0.3 | 0.5 | 0.3 | 0.1 | -0.6 | -1.0 | 0.0 | **-0.3** |
| NESM3 | -1.1 | -1.8 | -2.0 | -1.3 | -0.2 | 0.4 | 1.4 | 1.5 | 1.9 | 0.5 | -1.7 | -1.5 | **-0.3** |
| NorESM2-LM | 1.3 | -0.8 | -1.0 | -1.3 | -2.7 | -2.0 | 0.3 | 1.5 | 1.1 | 0.9 | 1.0 | 1.7 | **0.0** |
| NorESM2-MM | -0.3 | -1.0 | -0.8 | -1.8 | -2.6 | -1.3 | 0.6 | 0.4 | 0.6 | 0.7 | 0.5 | 0.4 | **-0.4** |

55

60

65

70 A list of the GCMs used is missing.

Yes, good point. We initially refrained from adding one to avoid comparisons between models, but we understand that this might be useful to many. To address this, we propose adding Table A1 in Appendix A, which details the GCMs used, along with their institutions and resolutions. Additionally, we will rename the existing appendix accordingly to maintain clarity and organization.

75 Table A1. List of GCMs along with their respective institutions and horizontal resolutions.

| GCM Name | Institution | Resolution (lat. x lon.) |
|---|---|---|
| *ACCESS-ESM1-5* | Australian Community Climate and Earth System Simulator, Australia | 1.25° x 1.875° |
| *CMCC-ESM2* | Centro Euro-Mediterraneo sui Cambiamenti Climatici, Italy | 0.9° x 1.23° |
| *CanESM5* | Canadian Centre for Climate Modelling and Analysis, Canada | 2.8° x 2.8° |
| *EC-Earth3* | | ~ 80 km |
| *EC-Earth3-CC* | European consortium of national meteorological services and research institutes; Spain, Denmark, Italy, Finland, Germany, Ireland, Portugal, Netherlands, Sweden, Norway, and Belgium. | ~ 80 km |
| *EC-Earth3-Veg-LR* | | ~ 125 km |
| *FGOALS-g3* | Chinese Academy of Sciences, China | 2.25° x 1.875° |
| *GFDL-ESM4* | Geophysical Fluid Dynamics Laboratory, USA | 1° x 1.25° |
| *INM-CM4-8* | | 1.5° x 2° |
| *INM-CM5-0* | Institute for Numerical Mathematics, Russia | 1.5° x 2° |
| *IPSL-CM6A-LR* | Institut Pierre Simon Laplace, France | 1.26° x 2.5° |
| *MIROC6* | Japan Agency for Marine-Earth Science and Technology, Japan | 1.4° x 1.4° |
| *MPI-ESM1-2-HR* | | 0.9375° x 0.9375° |
| *MPI-ESM1-2-LR* | Max Planck Institute for Meteorology, Germany | 1.875° x 1.875° |
| *MRI-ESM2-0* | Meteorological Research Institute, Japan | 1.125° x 1.125° |
| *NESM3* | Nanjing University of Information Science and Technology, China | 1.875° x 1.875° |
| *NorESM2-LM* | | 1.875° x 2.5° |
| *NorESM2-MM* | Norwegian Climate Centre, Norway | 0.9375° x 1.25° |

This analysis was probably already done by Talbot et al. (2024b). Still, it should also be shown in this paper, mainly because Talbot et al. (2024b), and even Talbot et al. (2024a), are still under review, and leaving some essential details in other papers still under review is not advisable (in other words, please take care that all the essential details for this study are within the
80 manuscript, and not delegated to other articles that, in principle, might not even be published).

We understand the importance of ensuring that all essential information for this study is included within the manuscript and not reliant on other unpublished papers. To address this concern, we will remove all citations to Talbot et al. (2024a) and Talbot et al. (2024b) where critical information is discussed and replace them with references to relevant published papers. Additionally, we will ensure that any critical details specific to this study are fully integrated into the manuscript to maintain
85 its completeness and self-sufficiency.

Furthermore, GCM resolution could be very rough for the small catchments subject of the study, especially considering the hydrological model resolution (1000 meters according to L162). What is the GCM resolution? How did the authors deal with the different resolutions of the GCMs and the hydrological model?

This is a valid concern about the resolution discrepancy between the GCMs and the finer grid of the hydrological model. Most CMIP6 models offer improved resolution compared to their CMIP5 counterparts. For example, the resolution of IPSL-CM5-LR (CMIP5) was $1.9° \times 3.75°$, while IPSL-CM6A-LR (CMIP6) improved to $1.25° \times 2.5°$. However, some models, such as CanESM2 and CanESM5, maintained a coarser resolution of $2.8° \times 2.8°$.

To address this mismatch, we applied the following approach:

1. For each catchment, values were extracted from the nearest GCM grid points.

2. The WaSiM hydrological model internally applies an inverse distance weighting (IDW) interpolation method for downscaling.

It is important to note that we deliberately avoided applying external downscaling methods for the conventional approach. This decision was made to ensure that both the conventional and asynchronous methods remain comparable, as introducing downscaling in only one method would create a methodological imbalance. Therefore, both methods use data at the same spatial scale as inputs.

Moreover, the results of our study are based on 30-year averages, which inherently smooth out year-to-year variability. Any additional precision gained from external downscaling of GCM data would likely be lost when averaged over such long periods. Thus, the coarser resolution of the GCMs is not expected to significantly impact the comparative analysis.

The primary goal of our study is to evaluate the general representation of hydrological processes by two methods, rather than to draw conclusions about the spatial variability of climate change impacts or specific events. Since no spatial analysis (e.g., variations due to soil types or land use) is performed, the precision of the meteorological input data at fine spatial scales is less critical.

Finally, this choice also simplifies the study by avoiding unnecessary complexity and uncertainty associated with selecting and applying a specific downscaling method. Such simplifications are particularly important for maintaining the methodological integrity of the asynchronous approach, which already involves unique calibration challenges.

We will clarify these points in the revised manuscript to ensure transparency regarding our methodological choices.

The same comment about spatial resolutions holds for the ERA5 dataset, which is preferred to ERA5-Land.

This comment is valid and very much in line with the previous. As such, our response is similar. The ERA5 dataset has a resolution of approximately 31 km, which provides multiple grid points per catchment. We applied the same methodology as with the GCMs, where the WaSiM hydrological model interpolated between ERA5 grid points using an inverse distance weighting (IDW) method to generate pixel-level data for precipitation and temperature.

While ERA5-Land offers a finer resolution of approximately 9 km, it is important to consider the nature of our study. The analysis focuses on 30-year averages, which inherently smooth out spatial and temporal variability. As a result, the additional spatial precision provided by ERA5-Land is unlikely to significantly alter the outcomes. Moreover, the study area is non-

120     mountainous, and the meteorological data showed minimal spatial variability between ERA5 stations. These factors suggest that using ERA5-Land would result in only marginal improvements in the resolution of hydrological estimates.

We acknowledge that using ERA5-Land could be an interesting avenue for future studies, particularly in regions with greater topographic variation or where finer spatial analyses are necessary. However, for the purposes of this study, the use of ERA5 represents a deliberate choice to simplify the methodology and focus on the comparative evaluation of hydrological modeling

125     approaches.

We will add a discussion of this point in section 2.2.1 to the revised manuscript to provide additional context for this methodological choice.

The last major concern regards the different objective functions used for the calibration of the models in the two approaches. I guess that the RMSE-based calibration supports the high flow calibration better (the same authors at LL614-617 somehow

130     admit this). Nevertheless, the problem of preserving peak flows could be solved with the conventional approach using other bias correction methods instead of (or in addition to) that chosen (e.g., a simple change factor method). In my opinion, the choice of the bias correction method and the objective functions limits the generalizability of the results achieved, and this should be at least discussed in more detail.

The reviewer correctly highlights that using RMSE as the calibration objective function in the asynchronous method prioritizes

135     high-flow events. We also acknowledge the concern regarding the representation of peak flows in the conventional approach. To further investigate the impact of the bias correction method, we conducted additional simulations comparing the multivariate bias correction (MBCn) method used in the study with the simpler delta (simple change factor) method. Figure R1 (provided below) presents the results of the conventional method with MBCn, the conventional method with the delta method, and the asynchronous method during the reference period, formatted similarly to Figure 4 in the manuscript. The

140     results indicate that the choice of bias correction method has minimal impact on the outcomes of the conventional method. The primary differences in results are linked to the modeling approach (asynchronous vs. conventional). Moreover, while the delta method shows greater variability across climate models compared to MBCn, its median values are closer to those of MBCn than to the asynchronous method in most cases. This suggests that the less effective preservation of peak flows in the conventional approach was not resolved by using the simpler delta method.

145     Our decision to use MBCn was driven by its ability to correct multiple variables simultaneously while preserving inter-variable dependencies, which are critical for physically consistent hydrological modeling. Simpler methods, like the delta or quantile mapping methods, fail to account for these inter-variable relationships, which are essential for accurately capturing the complex interactions between precipitation, temperature, and hydrological processes.

We agree with the reviewer that the choice of both the bias correction method and the calibration objective function influences

150     the results and their generalizability. To address this, we will expand the discussion in the revised manuscript to emphasize the implications of these methodological choices and acknowledge the limitations they impose on the broader applicability of the findings.

[Figure]

Figure R1. Comparison of the differences (%) in flow percentiles (a) Q95%, (b) Q90%, (c) Q50%, (d) Q10%, and (e) Q5% for 10 catchments under three methods: conventional using MBCn (blue), conventional using Delta (light blue), and asynchronous (orange). The differences are calculated relative to the observed flow.

Finally, please find below some other minor comments that should be addressed. I hope my review helps improve the quality of the paper.

We thank the reviewer for the detailed comments, which will definitely improve the clarity and quality of our manuscript. Below, we address each point and outline the corresponding revisions.

Title: I don't see that the paper moves towards a semi-asynchronous method. It looks more like a speculation in the discussion. Therefore, I suggest changing the title to something like "Comparison between conventional and asynchronous methods for…"

We acknowledge that the current title may overstate the development of a semi-asynchronous method, as it is primarily a speculation in the discussion. To better reflect the study's scope and focus, we propose to revise the title to something more descriptive: "Highlighting Challenges in Implementing the Asynchronous Method for Hydrological Modeling in Climate Change Impact Studies".

Abstract: it is unclear because the explanation of the asynchronous method is too concise and not exhaustive

We acknowledge that the explanation of the asynchronous method in the abstract is too concise. To address this, we will revise the abstract to include a clearer and more detailed explanation of the asynchronous method to ensure its concept and relevance are fully conveyed.

L59: "to reduce potential biases in the observed data" Do the authors mean "to reduce potential biases with respect to the observed data"?

The phrase "to reduce potential biases in the observed data" will be revised for clarity to: *"to reduce potential biases with respect to the observed data."*

LL63-64: as stated before, indeed, GCMs are the primary source of uncertainty. However, concerning BC's impact on hydrological variables, please consider 10.1016/j.ejrh.2022.101120 and references within and, more recently, 10.1016/j.ejrh.2024.101973

We will incorporate the recommended studies (10.1016/j.ejrh.2022.101120 and 10.1016/j.ejrh.2024.101973) into the manuscript while tackling the major comment above related to bias correction impacts.

L80: The explanation of the asynchronous method is not yet clear. What are the proxies?

We will clarify in the revised manuscript that proxies in the asynchronous method refer to statistical properties of observed streamflow, such as percentiles and distributions, which serve as calibration targets in this approach.

Table 1: please explain what it means that the annual rainfall is derived from a hydrological model

We will clarify in the manuscript that the annual rainfall values in Table 1 are derived from hydrological model simulations forced with ERA5 data. The hydrological model downscales the ERA5 data to a 1000 m × 1000 m resolution, and the values presented represent the average of all pixels across all catchments. Therefore, this is the precipitation at the catchment scale after processing, and not the raw data from ERA5 at its original resolution.

LL100-103: since two climate classes are named, it makes sense to see where they are on a map

To provide additional clarity, we will specify in the text that only the Godbout, Matane, and Bonaventure catchments belong to the Dfc climate class, while all other catchments are classified under the Dfb climate class.

L118: please explain why the ERA5-Land dataset was not considered for this study

As mentioned previously, ERA5 was used in this study, but it would be interesting to repeat the analysis using ERA5-Land, which has a higher spatial resolution of 9 km. The increased number of grid points per catchment could enable the hydrological model to produce more refined pixel estimates. However, since the study area is non-mountainous and the meteorological data showed minimal spatial variation between ERA5 stations, the potential improvement from using ERA5-Land would likely be minimal. We will acknowledge this limitation in the discussion section.

L236: here it is 1984-2011, but before (L226) it was 1984-2009.

We will correct the study period for consistency throughout the manuscript. The correct period is 1984-2011, and this adjustment will be reflected accordingly in L226.

200 L253: If I understand correctly, each catchment and climate model (therefore, 180 combinations) has its own parameter set, with its values for the parameters listed in Table 2. Please explain what the 1000 trials are

You are correct; each catchment and climate model combination (180 in total) has its own parameter set with values for the parameters listed in Table 2. To clarify, the term "1000 trials" refers to 1000 evaluations of the calibration algorithm. We will revise the text to replace "trial" with "evaluation" for greater clarity.

205 LL308-309: this outcome is not clear from Fig. 2

We will clarify in the text to ensure that the outcome presented in Figure 2 is clear to readers. The original sentence will be revised as follows:

Original:

"Despite these differences, the asynchronous method outperforms the conventional method in terms of annual volume accuracy
210 in 8 out of 10 catchments (Fig. 2)."

Proposed:

"Despite these differences, the asynchronous method outperforms the conventional method in terms of annual volume accuracy in 8 out of 10 catchments (Fig. 2). The annual volumes are indicated in the legend of each subplot, and in 8 out of 10 subplots, the asynchronous method's annual volume is closer to the observed values."

215 Fig.4 is adequate to support text in LL344-356. However, an additional table with some further statistics could help to make the analysis less qualitative (e.g., the sentence "When comparing the observed streamflow to the reference period simulations, the asynchronous method shows a closer alignment with the observed distribution": how much closer?)

We thank the reviewer for pointing this out. To address this, we propose adding five supplementary tables to the Appendix. These tables provide detailed statistical analyses of streamflow distributions (Q95%, Q90%, Q50%, Q10%, Q5%) across the
220 10 catchments for both the reference and future periods, using conventional and asynchronous methods.

These tables will include metrics such as the median (med) and standard deviation (std) of the streamflow distributions, providing a quantitative comparison to support the result of Figure 4.

The addition of these tables will make the analysis less qualitative by offering a clear, numerical assessment of the differences between methods and their ability to represent streamflow distributions under current and future conditions. Below are the
225 proposed tables:

Table C1. Analysis of Q95% streamflow distribution across 10 catchments for the reference and future periods using conventional and asynchronous methods. Metrics include the median (med) and standard deviation (std) of Q95% values (%).

| Catchment Name | Q95% | | | | | | | |
|---|---|---|---|---|---|---|---|---|
| | Reference period | | | | Future period | | | |
| | Conventional | | Asynchronous | | Conventional | | Asynchronous | |
| | med (%) | std (%) | med (%) | std (%) | med (%) | std (%) | med (%) | std (%) |
| Bonaventure | 4.5 | 4.1 | 0.3 | 4.1 | -15.3 | 9.2 | -15.6 | 8.2 |
| Matane | -4.2 | 3.2 | -0.3 | 4.1 | -21.8 | 11.3 | -10.5 | 9.6 |
| Ouelle | 6.8 | 3.3 | 1.6 | 3.0 | -12.0 | 10.5 | -7.8 | 13.6 |
| Bécancour | -11.7 | 3.1 | -0.6 | 3.4 | -17.7 | 5.8 | -7.9 | 7.7 |
| Nicolet SO | -3.4 | 2.7 | 4.1 | 7.2 | -20.7 | 6.0 | -6.2 | 9.5 |
| Au Saumon | 4.8 | 4.5 | 2.3 | 4.0 | -12.3 | 8.0 | -11.3 | 10.9 |
| Bras du Nord | -6.1 | 2.3 | 2.7 | 3.9 | -16.2 | 6.5 | -4.1 | 8.9 |
| Du Loup | 1.1 | 3.1 | -0.1 | 3.2 | -7.3 | 6.4 | -6.8 | 10.6 |
| Valin | -6.7 | 3.4 | 1.6 | 3.4 | -11.0 | 8.2 | -0.5 | 10.1 |
| Godbout | 2.5 | 3.9 | -3.0 | 3.9 | -5.1 | 9.5 | -4.9 | 10.1 |
| **Median** | **-1.2** | **3.3** | **0.9** | **3.9** | **-13.8** | **8.1** | **-7.3** | **9.8** |
| **Mean deviation** | **5.2** | **0.5** | **1.6** | **0.7** | **4.4** | **1.6** | **3.1** | **1.2** |

Table C2. Same as C1, but for Q90%.

| Catchment Name | Q90% | | | | | | | |
|---|---|---|---|---|---|---|---|---|
| | Reference period | | | | Future period | | | |
| | Conventional | | Asynchronous | | Conventional | | Asynchronous | |
| | med (%) | std (%) | med (%) | std (%) | med (%) | std (%) | med (%) | std (%) |
| Bonaventure | -4.1 | 3.3 | 4.9 | 4.9 | 1.1 | 9.6 | -8.1 | 8.5 |
| Matane | -10.8 | 2.2 | 1.8 | 4.9 | -10.0 | 11.2 | -9.6 | 10.1 |
| Ouelle | 5.1 | 3.9 | 3.1 | 4.3 | 0.8 | 10.6 | 0.3 | 12.6 |
| Bécancour | -8.1 | 3.0 | 1.0 | 4.3 | -3.9 | 6.9 | 0.9 | 8.5 |
| Nicolet SO | -2.5 | 2.5 | 3.7 | 5.9 | -9.9 | 6.9 | -3.3 | 10.0 |
| Au Saumon | 0.8 | 2.3 | -1.0 | 3.7 | -5.7 | 7.3 | -9.3 | 10.0 |
| Bras du Nord | -17.3 | 1.9 | 1.1 | 3.4 | -13.4 | 6.9 | 0.5 | 8.6 |
| Du Loup | -9.0 | 3.1 | 4.4 | 4.6 | 2.9 | 8.2 | -0.9 | 11.1 |
| Valin | -20.8 | 1.6 | 1.6 | 3.2 | -16.8 | 7.3 | -1.4 | 10.5 |
| Godbout | -11.9 | 2.4 | 2.2 | 3.2 | -6.2 | 9.5 | -0.1 | 10.1 |
| **Median** | **-8.5** | **2.4** | **2.0** | **4.3** | **-6.0** | **7.8** | **-1.2** | **10.1** |
| **Mean deviation** | **6.2** | **0.6** | **1.4** | **0.7** | **5.2** | **1.4** | **3.6** | **0.9** |

Table C3. Same as C1, but for Q50%.

| Catchment Name | Q50% | | | | | | | |
| --- | --- | --- | --- | --- | --- | --- | --- | --- |
| | Reference period | | | | Future period | | | |
| | Conventional | | Asynchronous | | Conventional | | Asynchronous | |
| | med (%) | std (%) | med (%) | std (%) | med (%) | std (%) | med (%) | std (%) |
| Bonaventure | 12.8 | 3.6 | 2.2 | 6.6 | 32.9 | 18.9 | 12.5 | 24.4 |
| Matane | -11.3 | 3.3 | -5.0 | 6.0 | 4.8 | 22.7 | 9.1 | 13.4 |
| Ouelle | -15.0 | 4.0 | -5.3 | 6.5 | 5.2 | 20.8 | 14.0 | 15.5 |
| Bécancour | -3.3 | 4.7 | -3.2 | 7.1 | 15.6 | 15.2 | 17.8 | 13.3 |
| Nicolet SO | -20.5 | 4.5 | -7.1 | 6.5 | -5.2 | 14.4 | 3.2 | 13.7 |
| Au Saumon | -9.0 | 4.5 | 6.1 | 7.1 | 3.7 | 12.4 | 10.4 | 18.3 |
| Bras du Nord | -13.9 | 2.5 | 13.6 | 4.3 | 1.9 | 13.6 | 27.8 | 16.8 |
| Du Loup | -1.5 | 2.6 | -2.1 | 10.0 | 15.9 | 15.9 | 11.7 | 14.4 |
| Valin | -23.4 | 2.3 | -1.0 | 7.9 | -5.5 | 13.7 | 12.0 | 15.2 |
| Godbout | 2.4 | 2.7 | 1.4 | 10.9 | 21.9 | 16.3 | 21.1 | 17.3 |
| **Median** | **-10.2** | **3.5** | **-1.5** | **6.8** | **5.0** | **15.5** | **12.3** | **15.3** |
| **Mean deviation** | **8.7** | **0.8** | **4.7** | **1.4** | **9.9** | **2.7** | **4.9** | **2.4** |

240

Table C4. Same as C1, but for Q10%.

| Catchment Name | Q10% | | | | | | | |
| --- | --- | --- | --- | --- | --- | --- | --- | --- |
| | Reference period | | | | Future period | | | |
| | Conventional | | Asynchronous | | Conventional | | Asynchronous | |
| | med (%) | std (%) | med (%) | std (%) | med (%) | std (%) | med (%) | std (%) |
| Bonaventure | 35.1 | 3.1 | 8.1 | 14.9 | -4.4 | 14.0 | -12.2 | 26.3 |
| Matane | -2.7 | 2.9 | 21.8 | 13.2 | -27.2 | 17.5 | 19.8 | 27.2 |
| Ouelle | 105.1 | 6.3 | 90.0 | 49.6 | 35.3 | 32.4 | 92.1 | 84.4 |
| Bécancour | 6.0 | 3.9 | 14.3 | 11.8 | -28.1 | 8.0 | 43.2 | 23.5 |
| Nicolet SO | 29.9 | 4.4 | 62.5 | 27.8 | -27.0 | 26.0 | 43.0 | 53.4 |
| Au Saumon | 38.0 | 4.6 | 22.7 | 27.3 | 13.2 | 25.2 | -1.6 | 45.8 |
| Bras du Nord | -0.3 | 2.2 | -30.4 | 15.6 | 4.6 | 12.0 | -37.5 | 36.3 |
| Du Loup | 89.4 | 3.9 | 34.6 | 12.2 | 82.4 | 14.2 | 52.2 | 35.2 |
| Valin | 31.8 | 1.5 | 17.4 | 10.4 | 33.9 | 9.2 | 2.3 | 28.1 |
| Godbout | 70.8 | 1.0 | 15.0 | 9.5 | 84.1 | 10.7 | 31.0 | 36.3 |
| **Median** | **33.4** | **3.5** | **19.6** | **14.1** | **8.9** | **14.1** | **25.4** | **35.8** |
| **Mean deviation** | **28.9** | **1.2** | **22.1** | **9.4** | **33.8** | **6.7** | **29.1** | **12.9** |

245

Table C5. Same as C1, but for Q5%.

| Catchment Name | Q5% | | | | | | | |
| | Reference period | | | | Future period | | | |
| | Conventional | | Asynchronous | | Conventional | | Asynchronous | |
| | med (%) | std (%) | med (%) | std (%) | med (%) | std (%) | med (%) | std (%) |
|---|---|---|---|---|---|---|---|---|
| Bonaventure | 39.3 | 5.4 | 12.0 | 15.3 | 2.2 | 11.3 | -8.5 | 29.7 |
| Matane | 4.9 | 3.9 | 34.2 | 17.6 | -30.1 | 13.8 | 23.8 | 34.2 |
| Ouelle | 158.4 | 12.7 | 137.6 | 81.3 | 55.2 | 28.6 | 125.2 | 113.6 |
| Bécancour | 26.5 | 5.7 | 42.5 | 18.5 | -8.3 | 6.9 | 78.5 | 29.9 |
| Nicolet SO | 66.1 | 8.1 | 109.9 | 48.1 | -25.4 | 25.0 | 73.6 | 74.6 |
| Au Saumon | 61.2 | 5.6 | 31.7 | 41.1 | 23.2 | 31.1 | 15.1 | 53.0 |
| Bras du Nord | 3.2 | 2.7 | -42.0 | 13.5 | 8.9 | 11.7 | -58.7 | 31.9 |
| Du Loup | 129.8 | 5.6 | 60.8 | 20.0 | 124.1 | 17.3 | 85.3 | 44.6 |
| Valin | 53.4 | 2.0 | 27.2 | 11.6 | 52.5 | 8.6 | 10.3 | 36.5 |
| Godbout | 85.3 | 1.0 | 17.3 | 10.8 | 97.1 | 9.7 | 34.5 | 37.2 |
| **Median** | **57.3** | **5.5** | **33.0** | **18.0** | **16.1** | **12.7** | **29.2** | **36.8** |
| **Mean deviation** | **37.7** | **2.3** | **35.8** | **17.4** | **41.8** | **7.3** | **42.2** | **19.1** |

250   L420: Indeed, the ETa peak is shifted forward (and is higher) with the conventional method, which could be significant for agricultural water resources management (e.g., irrigation).

It is correct that the ETa peak is shifted forward and is higher with the conventional method. We propose to add this sentence at the end of the paragraph at line 420 to highlight the impacts on water use:

"*However, with the conventional method, the ETa peak is higher and occurs several weeks later than with the asynchronous* 255   *method, which could have significant implications for agricultural water resource management, particularly in irrigation planning.*"

L568: please consider that the results achieved could not be generalized but are specific for the considered catchment, including local climatology

While it is true that the study was performed on a particular region, the catchments cover a quite large region and are general 260   across these catchments, which supports generalizability within similar regions. The theoretical aspects also support the fact that results should be generalizable, notably with respect to the limitations of asynchronous modelling. We will however clarify this in the text.

The conclusions are a bit repetitive and don't add too much. They look more like a summary. Maybe discussion and conclusions could be merged.

265   To address this, we will review and revise the conclusion to reduce redundancy, making it more concise and focused on key takeaways. However, we do not plan to merge the conclusions with the discussion, as the journal's guidelines recommend keeping these sections distinct.

We greatly appreciate the reviewer's insightful comments, which have helped refine our study and improve the manuscript.

Sincerely,

270    Frédéric Talbot on behalf of all authors

---

## Author Comment (AC2)

**Towards a semi-asynchronous method for hydrological modeling in climate change studies**

Frédéric Talbot[1], Simon Ricard[3], Jean-Daniel Sylvain[2], Guillaume Drolet[2], Annie Poulin[1], Jean-Luc Martel[1], Richard Arsenault[1]

[1] Hydrology, Climate and Climate Change Laboratory, École de technologie supérieure, Université du Québec, Montréal, H3C 1K3, Canada

[2] Direction de la recherche forestière, Ministère des Ressources naturelles et des Forêts, Québec, G1P 3W8, Canada

[3] Sciences and Engineering, Université Laval, Québec, G1V 0A6, Canada

*Correspondence to*: Frédéric Talbot (frederic.talbot.2@ens.etsmtl.ca)

**RC2 (https://doi.org/10.5194/egusphere-2024-3037-RC2)**

Dear Editor,

The manuscript was well taken care -off and is tidy. However, I think the justification of the used metric for calibrating the hydrological model and not investigating its effect on the outcome I see as major shortcomings of this manuscript.

Review "Towards a semi-asynchronous method for hydrological modeling in climate change studies"

**General**

The manuscript compares two methods for deriving climate discharge scenarios. One with model calibration based on observed meteorological data + bias correction (conventional) and the other without bias correction and a hydrological model trained on climate model output (asynchronous) and optimized on statical metrics. Its compares the performance of the conventional and asynchronous method in simulating hydrological processes under both current and future climate condition.

We sincerely appreciate RC2 time and effort in reviewing our manuscript. We are glad that RC2 considers that our text was carefully prepared and is well-organized. We have indeed spent much time thinking about how to present our work in the most complete and understandable manner. We appreciate RC2 constructive feedback and acknowledge the concern regarding the justification of the calibration metric and its potential impact on the study's outcomes. Below, we provide a detailed response and outline the revisions we will make to address this issue.

**Main comments**

In general the manuscript reads well and setup of the research is clear and and well described. However, It stays unclear what the semi-aynschronous approach is.

The manuscript basically investigates both approaches which makes the additional value (to the available literature) unclear. The main conclusions are defined by the choice of the optimization metric (Kling KGE in this conventional case) and the selected stat. metric (RMSE of the sorted discharges) in the asynchronous method. The choice of this metric is not supported by any reasoning/evidence while it is clear this defines the results and any conclusion. It would be relatively easy to define and include more (better?) statistical metrics (also to make it fairer in comparison with the metric for the conventional

approach) that take into account timing, magnitude etc. This needs much more attention/work in the manuscript. Why was this metric selected and not other ones?

35 We thank the reviewer for this comment, as it reveals that we did not provide a clear enough explanation of the method. The asynchronous method is a relatively new addition to the hydrological modeling field, first introduced in 2019. While it has been applied in a few studies to assess the impacts of climate change, a systematic hydrological variables comparison with the widely used conventional approach, using a physically-based hydrological model, had not been undertaken prior to this work. This study provides a novel contribution by focusing on the representation of hydrological variables under the asynchronous

40 method and cautioning researchers and practitioners about its current limitations. Specifically, while the asynchronous method may achieve seemingly accurate streamflow results, the mechanisms underlying these results often fail to align with hydrological processes, leading to inconsistent representation of variables like snowmelt, which reduces the level of confidence that can be attributed to it under future climate scenarios. These findings underscore the need for significant improvements to the method to make it reliable for climate change impact assessments. Furthermore, the analysis of intermediate variables

45 highlights the issue of synchronicity, which is currently overlooked by the asynchronous method. This underscores the necessity of introducing an objective function that incorporates the concept of synchronicity, thereby paving the way for the development of a semi-synchronous approach. The semi-synchronous approach, as introduced in the discussion section, would retain the distributional benefits of the asynchronous method while addressing the timing discrepancies by aligning hydrological processes more closely with observed seasonal dynamics.

50 For the comment regarding the calibration metrics, for the conventional method, we employed the widely accepted Kling-Gupta Efficiency (KGE), which evaluates model performance by balancing correlation, variability, and bias. This metric is a standard in hydrological modeling and aligns with best practices for the conventional approach.

For the asynchronous method, we calibrated against the distribution of streamflow using raw GCM data, which inherently lacks temporal synchronization with observed flows. This is the crux of the asynchronous method, as the flows are not

55 synchronized between GCM-driven simulated and observed flows. Therefore, we required an objective function that does not rely on timing. Root Mean Square Error (RMSE) on the distribution of flows was chosen as it is well-established, simple, and excludes correlation in timing, making it suitable for this purpose.

We agree with the reviewer that other metrics excluding correlation could also be considered and might yield different results. However, our goal was to evaluate the current state of the asynchronous method rather than proposing a modified approach.

60 In the discussion section of the manuscript, we propose integrating metrics that incorporate some form of timing (i.e. seasonal) to address issues related to the misrepresentation of hydrological processes.

To further clarify the asynchronous method's workflow, we propose to include the below workflow diagram in Section 2.3.3 of the revised manuscript. This diagram is adapted from Ricard et al. (2023, https://doi.org/10.5194/hess-27-2375-2023).

We will also expand the discussion to explicitly address the implications of using alternative metrics for the asynchronous

65 method.

[Figure]

Figure 2. Workflow diagram of the asynchronous method.

Moreover it remains questionable what the value is of calibrating a hydrological model on poor meteorological forcing.

It is important to note that the asynchronous method is an existing approach in hydrological modeling. Its added value lies in its ability to function without requiring meteorological observations or a bias correction step, making it easier to implement compared to the conventional approach.

Additionally, the asynchronous method inherently preserves trends and maintains the physics of the GCM and consistency between simulated climate variables due to the absence of bias correction. This feature ensures that the original GCM signals are retained, which is particularly advantageous for capturing climate change impacts.

The objective of this study is not to propose or advocate for the asynchronous method but to critically evaluate its performance relative to the conventional method and to identify potential solution pathways. By systematically comparing these two existing approaches, this study aims to provide insights into their respective strengths, limitations, and suitability for different hydrological modeling scenarios.

Another approach would have been to use models that tune hyperparameters and are more focused on patterns / seamless parameters like the MPR (Samaniego et al. 2010) or follow a same approach even without calibration (example in this context is in Sperna Weiland et al., 2020). Something that could also have been used for the Wasim model used?

We appreciate the reviewer's suggestion to explore alternative approaches, such as using models like MPR (Samaniego et al., 2010). These approaches offer valuable insights into parameter transferability across scales and regionalization.

While our study does not focus on these aspects, we agree that applying multiscale parameter approaches to WaSiM could be an interesting direction for future research, and we will add the reference in the discussion. Such an investigation could enhance our understanding of parameter behavior across scales and further improve the robustness of hydrological modeling in this context.

90 **Refs:**

https://agupubs.onlinelibrary.wiley.com/doi/full/10.1029/2008WR007327

https://www.frontiersin.org/journals/water/articles/10.3389/frwa.2021.713537/full

**Specific comments**

563, 592 => another or more stat metrics could have minimized this effect. Why isn't this discussed

95 This is a good point. The potential use of additional statistical metrics to minimize the asynchronous method's challenges with synchronization is addressed in section 4.3 of the manuscript, where we discuss the potential improvement of the asynchronous method through the integration of synchronicity. Specifically, we propose the development of a semi-asynchronous approach that combines the strengths of both conventional and asynchronous methods.

The relevant passage (lines 649–657) states:

100 "Looking forward, one of the most promising avenues for improving the asynchronous method is the integration of synchronicity, leading to the development of a semi-asynchronous approach. This hybrid method would combine the strengths of both the conventional and asynchronous methods, offering a more balanced solution that mitigates the weaknesses observed in each. By incorporating synchronicity into the calibration process, the semi-asynchronous method would better align the timing of hydrological events, such as snowmelt, with observed data, improving its ability to capture critical seasonal

105 dynamics.

For instance, modifying the objective function to calibrate based on seasonal or monthly data could enhance the model's ability to simulate hydrological processes. This integration of event timing into the calibration process is crucial for addressing the timing discrepancies that currently limit the asynchronous method's performance."

This discussion highlights the importance of exploring alternative metrics, such as those focusing on seasonal or event-specific

110 timing, to improve (or partly integrate) synchronization in the asynchronous method. We will ensure this point is emphasized further in the revised manuscript to address the reviewer's concern.

665=> I see that you mention this here but I would have expected a thorough discussion in the discussion section and more in the material/methods section about this choice.

After reviewing line 665 and the surrounding sentences, we are unsure about which specific choice is being referenced. If the

115 comment pertains to the choice of calibration metrics, modeling approach, or another methodological aspect, we kindly ask for clarification to address the concern more directly.

We appreciate the valuable feedback and believe these revisions will significantly improve the clarity and scientific contribution of our study.

Sincerely,

120 Frédéric Talbot on behalf of all authors